# Impact of early life antibiotic and probiotic treatment on gut microbiome and resistome of very-low-birth-weight preterm infants

Raymond Kiu [1,2,3] ✉, Elizabeth M. Darby [1,2,9], Cristina Alcon-Giner[3,9], Antia Acuna-Gonzalez[3,9], Anny Camargo[3,4,5], Lisa E. Lamberte[2], Sarah Phillips[3], Kathleen Sim[6], Alexander G. Shaw[6], Paul Clarke [7,8], Willem van Schaik [1,2], J. Simon Kroll[6] & Lindsay J. Hall [1,2,3,8] ✉

Preterm infants (<37 weeks' gestation) are commonly given broad-spectrum antibiotics due to their risk of severe conditions like necrotising enterocolitis and sepsis. However, antibiotics can disrupt early-life gut microbiota development, potentially impairing gut immunity and colonisation resistance. Probiotics (e.g., certain *Bifidobacterium* strains) may help restore a healthy gut microbiota. In this study, we investigated the effects of probiotics and antibiotics on the gut microbiome and resistome in two unique cohorts of 34 very-low-birth-weight, human-milk-fed preterm infants - one of which received probiotics. Within each group, some infants received antibiotics (benzylpenicillin and/or gentamicin), while others did not. Using shotgun metagenomic sequencing on 92 longitudinal faecal samples, we reconstructed >300 metagenome-assembled genomes and obtained ~90 isolate genomes via targeted culturomics, allowing strain-level analysis. We also assessed ex vivo horizontal gene transfer (HGT) capacity of multidrug-resistant (MDR) *Enterococcus* using neonatal gut models. Here we show that probiotic supplementation significantly reduced antibiotic resistance gene prevalence, MDR pathogen load, and restored typical early-life microbiota profile. However, persistent MDR pathogens like *Enterococcus*, with high HGT potential, underscore the need for continued surveillance. Our findings underscore the complex interplay between antibiotics, probiotics, and HGT in shaping the neonatal microbiome and support further research into probiotics for antimicrobial stewardship in preterm populations.

The World Health Organisation (WHO) estimates that over 10% of infants are born prematurely each year worldwide, defined as gestation of <37 weeks[1]. Among newborns, Very Low Birth Weight (VLBW) infants, those born weighing below 1500 g, represent around 1.1–1.4%[2,3]. VLBW preterm infants have underdeveloped immune systems, making them particularly susceptible to morbidities such as Necrotising Enterocolitis (NEC)[4,5] and sepsis[6,7], often

involving antibiotic-resistant bacteria. Due to these risks, preterm infants are frequently cared for in Neonatal Intensive Care Unit (NICUs) and are routinely administered broad-spectrum antibiotics, typically benzylpenicillin and gentamicin or derivatives, during their first days and weeks of life[8,9]. However, this early-life antibiotic exposure can disrupt the normal development of the gut microbiota[8,10,11].

---

The use of antibiotics in preterm infants can also lead to an enrichment of antibiotic resistance genes (ARGs), collectively known as the gut resistome[12]. ARGs within gut bacterial communities can spread rapidly, primarily through horizontal gene transfer (HGT), occurring both within and between species, including from commensals to pathogens or vice versa, and this transfer often occurs via mobile genetic elements such as plasmids[13]. Multidrug-resistant bacteria, including *Staphylococcus*, *Klebsiella*, *Enterococcus* and *Escherichia* are common in the preterm infant gut[5,8,14], with their presence frequently linked to prolonged hospitalisation[15], late-onset bloodstream infections[16], and nosocomial infections in hospital settings[17–19].

In response to these challenges, WHO has recommended probiotic supplementation for very preterm (<32 weeks' gestation), human-milk-fed infants[20]. Probiotics, particularly *Bifidobacterium* spp. and *Lactobacillus* spp., are now increasingly used in NICUs[21], with ~40% of NICUs in the UK adopting this practice[8,9]. Probiotic supplementation has been associated with reduced NEC incidence, lower mortality, fewer gut pathogens, and enhanced immune maturation[8,14,21–28]. Importantly, probiotics have also been observed to reduce the abundance of ARGs in the gut microbiota of preterm infants, bringing it closer to levels seen in full-term infants[29–32].

In this work, we studied two cohorts of VLBW preterm infants, all exclusively fed with human milk to investigate the impact of probiotics on the preterm gut microbiome and resistome. Using shotgun metagenomics and genome-resolved approaches, we assessed microbiome species and strain dynamics, and the effects of probiotics and antibiotics during the first three weeks of life. Additionally, we performed a plasmid transfer experiment using *Enterococcus* in an infant gut model to explore ex vivo ARG transfer. Importantly, we demonstrate that probiotic supplementation significantly reduced the prevalence of ARGs and multidrug resistance (MDR) pathogens, while also aiding in the restoration of a typical early-life gut microbiota in preterm infants. In addition, the persistence of MDR pathogens such as *Enterococcus* in preterm gut, with significant horizontal gene transfer potential, underscores the need for ongoing clinical surveillance.

## Results

We analysed the gut microbiome from 34 VLBW preterm infants (moderate to very preterm), divided into two main cohorts: the Probiotic-Supplemented (PS) cohort and the Non-Probiotic-Supplemented (NPS) cohort. Infants in the PS cohort received probiotics containing *Bifidobacterium bifidum* and *Lactobacillus acidophilus*, while those in the NPS cohort did not receive any probiotics. These infants were selected as a subset from a larger study[8]. Within each cohort, some infants received empirical antibiotic treatment with benzylpenicillin and gentamicin, while others served as controls without antibiotic exposure (Fig. 1a). Faecal samples were collected weekly from preterm infants during the first three weeks of life, when possible (Supplementary Table 1). These samples underwent processing, sequencing, and computational analysis to characterise the gut microbiome.

### Probiotics suppress pathobionts and support restoration of the early life gut microbiota

Gut microbiome diversity differed significantly between the NPS and PS cohorts, with the NPS cohort showing an over-time increase in microbiome diversity and genus/species richness, while the PS cohort maintained similar levels of diversity throughout (Fig. 1b, Supplementary Fig. 1a, b). The overall gut microbiome profiles were markedly distinct between the NPS and PS groups, with *Bifidobacterium* identified as key driving taxon for PS cohort (Fig. 1c). The NPS infant gut microbiomes were characterised at the genus level by early-life pathobionts (Fig. 1d), including *Klebsiella*, *Enterobacter*, *Escherichia*, *Enterococcus*, and *Staphylococcus*, which on the species level were identified as *Klebsiella pneumoniae*, *Klebsiella grimontii*, *Enterobacter*

*hormaechei*, *Escherichia coli*, *Staphylococcus epidermidis*, and *Staphylococcus haemolyticus* (Fig. 1e). In contrast, the gut microbiomes of PS infants were dominated by the genus *Bifidobacterium*, particularly *B. bifidum*, a major component of the Infloran probiotic provided to the infants, exhibited active replication (Index of Replication >1.5) in the preterm gut, highlighting the impact of probiotic supplementation (Fig. 1f). Notably, *Bifidobacterium breve* and *Bifidobacterium longum*, both associated with breastfeeding and recognised for promoting a healthy infant gut by breaking down complex carbohydrates, including human milk oligosaccharides (HMOs), appeared earlier and were more abundant in PS infants compared to NPS infants (Fig. 1g).

Comparative analysis showed that *Bifidobacterium* was significantly more abundant in the PS cohort microbiota, whereas *Enterobacter*, *Escherichia*, *Klebsiella* and *Veillonella* were less abundant (Fig. 1h, Supplementary Fig. 2a, b). This was supported by read coverage data, which showed a consistent reduction in *Klebsiella* and *Escherichia* (Supplementary Fig. 2c, d). *Staphylococcus* was notably dominant only during the first week of life and declined in both cohorts by weeks 2 and 3, indicating its transient presence in the infant gut. Importantly, *Enterococcus*, a frequently multidrug-resistant genus including *Enterococcus faecalis*, was also a prominent member of the preterm gut microbiota in both cohorts.

Next, we examined the functional pathways within the gut microbiomes. Clustering of these pathways showed increasingly distinct patterns between the PS and NPS cohorts starting from week 2 (Fig. 1i). Across all samples, a total of 407 functional pathways were identified, with 84 pathways showing significantly different abundances between the two cohorts (Fig. 1j). Among these, 14 pathways (3.4%) were unique to the PS cohort, including sucrose degradation pathways, while 370 pathways (90.9%) were shared between both cohorts. In addition, 23 pathways (5.7%) were exclusively associated with NPS infants.

### Probiotic intervention is linked to suppression of ARGs in the gut resistome

We investigated the gut resistome in both NPS and PS cohorts, comprising infants treated empirically with benzylpenicillin and/or gentamicin, to assess the impact of these antibiotics on early gut microbiomes and resistomes over the first 3 weeks of life (Fig. 2a). Notably, the abundance of ARGs was significantly higher in NPS infants compared to PS infants across the first 3 weeks (Fig. 2b). Analysis of ARG diversity, defined by the number of antibiotic/drug classes represented, showed that NPS infant guts contained significantly more ARG types than those in the PS cohort, particularly in weeks 2 and 3 (Fig. 2c). Key ARG types in both cohorts included genes conferring resistance to aminoglycosides, macrolides-lincosamides-streptogramines (MLS), beta-lactamases, trimethoprim, and tetracycline, while ARGs conferring resistance to fluoroquinolones and colistin were exclusively found in NPS infants (Fig. 2d). Clustering analysis of resistome profiles revealed shared ARGs in both cohorts (48 ARGs), with 46 unique ARGs in NPS infants and 11 in PS infants, though the separation between groups was not clearly distinct (Fig. 2e, f).

To examine the impact of empirical antibiotic therapy on the gut microbiome, we compared Antibiotic-treated vs Control (non-antibiotic-treated) groups within each cohort. Overall, no significant difference in microbiome diversity was observed between Antibiotic and Control groups (Fig. 2g). In the NPS cohort, the Antibiotic group showed a more varied gut microbiome from the first week, which continued through weeks 2 and 3, while the Control group's microbiome became more complex in later weeks; both groups were dominated by key pathobionts (Fig. 2h). In the PS cohort, apart from an initial increase in *Staphylococcus* in Antibiotic-treated infants and *Enterococcus* in Controls, *Bifidobacterium* dominated in weeks 2 and 3. Taxonomic abundance analysis revealed no significant overall differences in abundance for Antibiotic vs Control in the NPS cohort, except

 

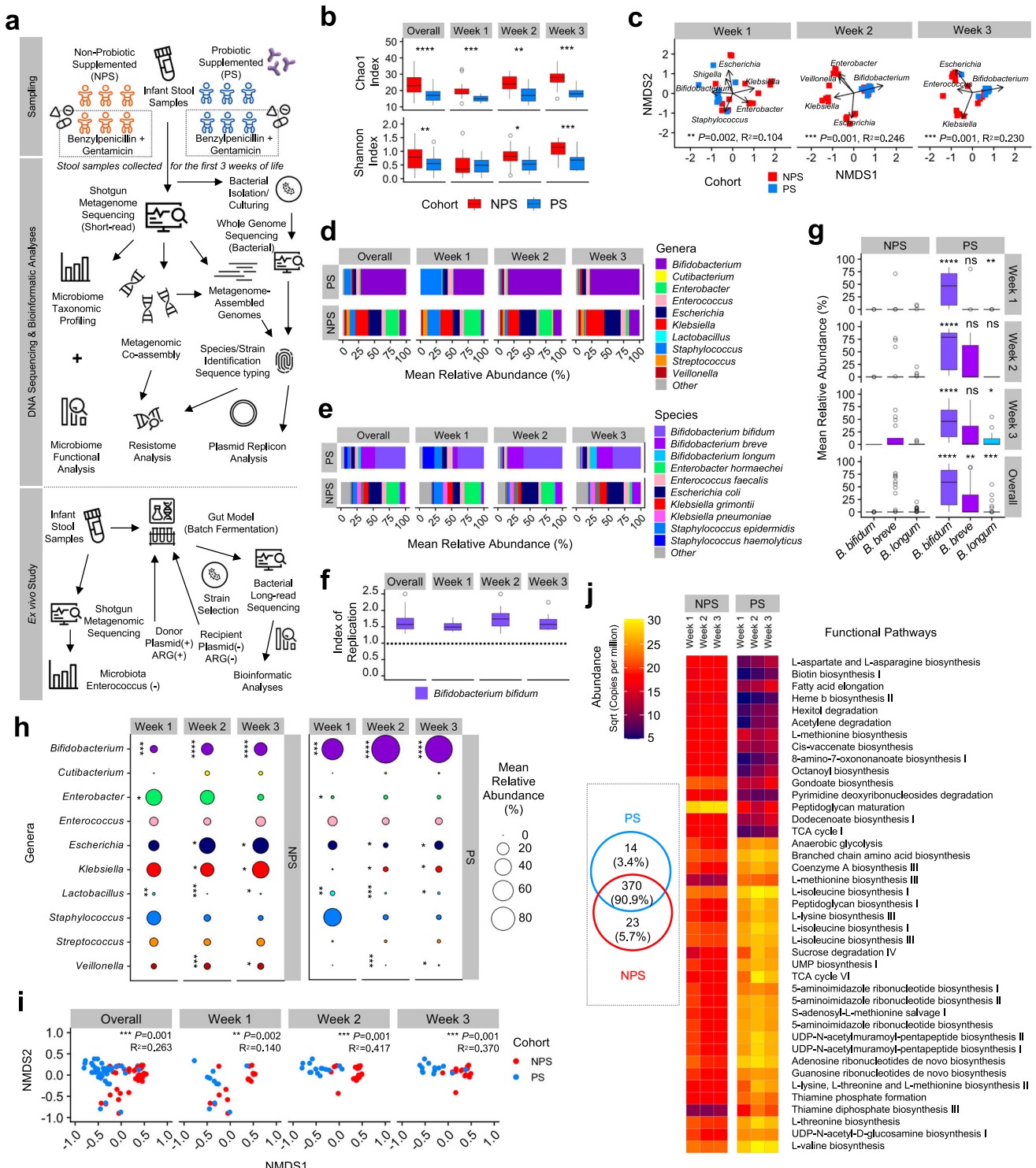

for *Bifidobacterium* and *Lactobacillus* (Fig. 2i, top). In the PS cohort, Antibiotic-treated infants had significantly higher levels of *Cutibacterium*, *Escherichia*, and *Lactobacillus*, with reduced *Enterococcus* and *Streptococcus* over 3 weeks (Fig. 2i, bottom). A further intra-cohort longitudinal analysis of antibiotic-exposed infants in both cohorts revealed an increase in *Bifidobacterium* abundance in the PS cohort by week 2 (compared to week 1), while overall *Staphylococcus* levels decreased across both cohorts in week 2 (Supplementary Fig. 3).

To further understand the impact of antibiotics on the resistome, we compared ARG profiles in Antibiotic vs Control groups within each cohort. No significant differences were observed in ARG counts

between these groups in either cohort (Fig. 2j). However, both Antibiotic-treated and Control infants in the NPS cohort had higher ARG counts than their counterparts in the PS cohort, suggesting a potential ARG-suppressive effect of *B. bifidum* in the preterm gut (Fig. 2k).

We also assessed whether microbiome diversity correlated with ARG abundance in preterm infants. Statistical testing indicated a moderate, but statistically significant, positive correlation between increased microbiome diversity and ARG levels in PS cohort (correlation coefficient $R = 0.36$; $P = 0.0019$) but not NPS (Fig. 2l). Importantly, analysis of the correlation between taxa abundance in gut

**Fig. 1 | Preterm infant gut microbiome analysis and functional profiling.**
**a** Schematic overview of the study design and analyses. **b** Gut microbiome genus richness (top) and diversity indices (bottom) for 34 infants across two cohorts ($n = 19$ NPS, $n = 15$ PS). **c** NMDS plots of gut metagenomes from 34 preterm infants during the first 3 weeks of life. Differences in microbiota composition between cohorts were tested using PERMANOVA, based on Bray-Curtis distances. The significance ($p$-values) and the proportion of variance explained ($R^2$) were shown. **d** Taxonomic relative abundance (%) of 34 infants at genus level, showing only the top 10 most abundant genera across all samples. **e** Species-level gut microbiome profiles of the same 34 infants, showing bacterial species from the top 10 most abundant genera. **f** iRep (Index of Replication) of probiotic *B. bifidum* Infloran strain in the PS cohort. Replication index >1.0 (dashed line) indicates active replication; >1.5 indicates rapid replication. ($n = 36$ PS samples) **g** Comparison of relative abundances (%) of three key *Bifidobacterium* species (where *B. bifidum* is a key component of probiotic supplementation) between the two cohorts ($n = 53$ NPS vs $n = 39$ PS), stratified by week. Mean relative abundances are shown. **h** Mean proportion of the top 10 most abundant bacterial genera in both cohorts displayed in bubble plots. **i** NMDS plots of gut microbiome functional profiles for 34 preterm infants, stratified by week and cohort (NPS vs PS). Differences in functional composition between cohorts were tested using PERMANOVA, based on Bray–Curtis distances. The significance ($p$-values) and the proportion of variance explained ($R^2$) were shown. **j** Heatmap of 41 significant functional pathways (alpha = 0.05, LDA score >2.0). Inset shows a Venn diagram depicting shared and unique pathways between NPS and PS cohorts based on 407 computationally identified pathways. In (**b**, **f** and **g**), the box plots represent median (line inside the box), interquartile range (IQR; middle 50% of the data, box height), data within 1.5 × IQR (whiskers), and outliers (points). In (**b**, **g** and **h**), statistical significances were assessed using two-sided Wilcoxon tests with Benjamini-Hochberg adjustment for multiple comparisons. Statistical significance: *$P < 0.05$, **$P < 0.01$, ***$P < 0.001$, ****$P < 0.0001$; ns not significant.

microbiomes and ARG count revealed a negative association with *Bifidobacterium*, whereas *Enterococcus* and *Staphylococcus* demonstrated positive correlations (Fig. 2m). In addition, a colistin resistance gene, *mcr-9.1*, was detected with 100% nucleotide identity in the gut microbiome of a preterm infant around 2011–2012 (Fig. 2n), predating its identification as a resistance gene in 2019, though it could not be computationally mapped to a specific bacterial strain in this study.

## Strain-level gut-derived pathobiont resistome analysis
To examine gut resistomes at the strain level, we applied genome-resolved metagenomics, recovering 322 high-quality metagenome-assembled genomes (MAGs) from shotgun metagenome sequences and incorporating 89 novel isolate genomes cultured from preterm infant faecal samples (Supplementary Fig. 4a–d). Together, these genomes represent 27 bacterial genera and 47 species. Resistance-gene searches and statistical analyses were conducted on these bacterial genomes to characterise the early-life gut resistomes.

Following dereplication of all genomes, including MAGs, we obtained 195 representative strains for further resistome analysis (Fig. 3a). In total, 10 ARG resistance classes (antibiotic classes) were identified across bacterial strains, with *Enterococcus*, *Escherichia*, *Klebsiella*, and *Staphylococcus* identified as the four most resistant genera, each exhibiting a significantly higher ARG count per strain vs the rest (Fig. 3b). Further examination of these genera showed a trend towards lower ARG counts in *Enterococcus*, *Escherichia*, and *Klebsiella* strains within the PS cohort compared to NPS, though differences were not statistically significant (Fig. 3c).

A deeper analysis of MDR efflux pumps across these resistant genera revealed multiple efflux pumps, with *Escherichia* and *Klebsiella* strains encoding the highest number ($n = 13$), followed by *Staphylococcus* ($n = 7$) and *Enterococcus* ($n = 6$) (Fig. 3d). ARG resistance class analysis further demonstrated the widespread presence of beta-lactamase resistance genes in *Escherichia*, *Klebsiella*, and *Staphylococcus* strains (Fig. 3e, top). Analysis of MDR capacity showed that NPS *Enterococcus* ranked highest among these resistant genera, with over half of its strains harbouring ARGs conferring resistance to more than three antibiotic classes, followed by NPS *Staphylococcus*. Notably, none of the PS-associated *Klebsiella* ($n = 5$) or *Escherichia* ($n = 12$) genomes were MDR, while 47.6% of NPS-associated *Escherichia* ($n = 31$) genomes, including cultured isolate genomes, exhibited MDR characteristics (Fig. 3e, bottom).

To assess clinical relevance, we sequence-typed a total of 193 key resistant pathobiont genomes from *Escherichia*, *Klebsiella*, *Enterococcus*, and *Staphylococcus* (Fig. 3f) using the multi-locus sequence typing (MLST)[33] scheme. Dominant sequence types (STs) included ST1193, ST127, ST394, and ST681 for *E. coli*, while ST432 was prominent for *K. pneumoniae*. In *Enterococcus*, 12 STs were identified, including six potential novel STs. *S. epidermidis* displayed 14 STs, with ST32 in 10 genomes and 12 potential novel STs. Of particular note, key *S.*

*haemolyticus* STs included methicillin-resistant ST1, linked to NICUs previously[34] and ST49, associated with children and often resistant to methicillin[35,36]. Notably, certain sequence types (STs) of *E. faecalis* (ST179 and ST40), *S. epidermidis* (ST110, ST153 and ST32) and *S. haemolyticus* (ST1 and ST49) were shared between both the NPS and PS cohorts, indicating clinical relevance. We also performed surface antigen typing for *Klebsiella*, identifying KL10 ($n = 10$) as the most common capsule (K) type and O1/O2v1 ($n = 15$) as the most prevalent LPS (O) type in the *Klebsiella* genomes (Fig. 3f, right).

## Mobilome analysis and circulation of strains within NICUs
To investigate whether plasmid carriage is linked to the resistome, we performed a plasmid replicon search across 195 strain-level bacterial genomes as well as metagenome level (Fig. 4a). Comparison between PS and NPS cohorts revealed significantly higher plasmid counts in the gut microbiomes within NPS cohort (Fig. 4b). To further assess the role of antibiotics in the mobilome, we compared plasmid counts between Antibiotic-treated and Control groups within both cohorts. Although Antibiotic-treated infant gut microbiomes showed a higher plasmid count, the difference was not statistically significant in either cohort (Fig. 4c) or in the overall comparison (Fig. 4d).

Next, we examined the correlation between plasmid replicon count and ARG count (metagenome level), finding a weak positive correlation overall ($\tau = 0.33$; $P = 0.00001$), however, signals were stronger in Antibiotic-treated infants vs Control ($\tau = 0.43$, $P < 0.0001$ vs $\tau = 0.20$, $P = 0.1998$; Fig. 4e). Notably, the Antibiotic-treated group exhibited a significantly higher frequency of potential horizontal gene transfer events compared to the Control group (Fig. 4f). Among Top 10 most abundant genera, *Enterococcus*, *Escherichia*, and *Staphylococcus* were identified as top plasmid carriers, with *Enterococcus* ranking highest in both ARG and plasmid carriage (Fig. 4g). Given its prominence as an MDR pathogen, its high prevalence of plasmid carriage, and its relative lack of study in preterm infants, we next focused on tracking *Enterococcus* transmission among infants in both cohorts.

Average nucleotide identity (ANI) analysis at a strain-level cut-off of 99.9% indicated 6 *Enterococcus* strains were circulating among infants, with four pairs of infants (unrelated) in Hospital A (one of which was a twin) and one pair in Hospital B carrying identical strains (Fig. 4h). Additionally, in Hospital B, three other unrelated infants were found to be colonised with the same *Enterococcus* strain, underscoring the potential for nosocomial transmission of this MDR pathogen.

## AMR plasmid transfer between *Enterococcus* strains occurs within an infant gut model
Given the clinical significance of *Enterococcus faecium* as a multidrug-resistant (MDR) pathogen and its relatively limited study in infant microbiomes, we hypothesised that ARG-encoding plasmids could be transferred between *E. faecium* strains within the infant gut. To test

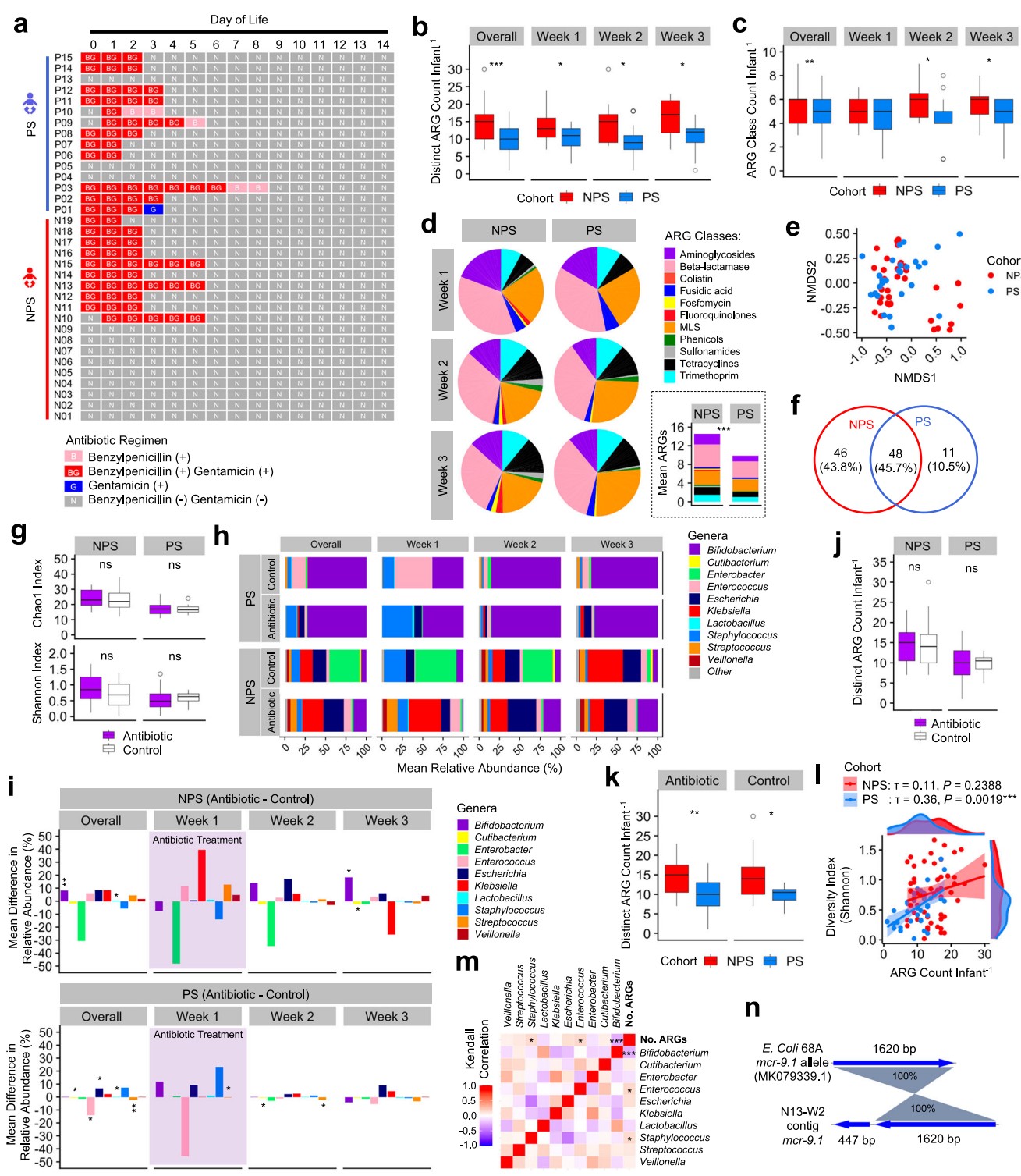

this, we used two donor strains, both originated from preterm infants, and a plasmid-free, gentamicin-sensitive recipient strain (64/3) in a plasmid transfer experiment (Fig. 5a). We utilised a gut model using *Enterococcus*-free infant faecal slurry as the culture medium to simulate the neonatal gut environment (Fig. 5b). Following 24 h incubation, we isolated *Enterococcus* colonies on selective media and performed Whole Genome Sequencing (WGS). Genome analysis revealed the successful transfer of a ~137 kb mega-plasmid encoding the aminoglycoside resistance gene *aac6-aph2* to plasmid-free recipient strain 64/3 (transformants D1 and D3), thereby conferring gentamicin resistance to these strains (Fig. 5c, d).

## Discussion

In this study, we used genome-resolved metagenomics to analyse the resistome and microbiome of preterm infants during their first 3 weeks of life in neonatal units, highlighting the complex effects of antibiotics, probiotics, and HGT on the gut microbiome. Our findings suggest that probiotic supplementation with *B. bifidum* and *L. acidophilus* not only supports beneficial microbial communities but also plays a role in reducing MDR bacteria and overall ARG carriage.

One of the strengths of this study is the inclusion of both PS and NPS infants who did not receive antibiotics, which is uncommon considering the high antibiotic prescription rates among these at-risk

**Fig. 2 | In-depth profiling of the neonatal preterm infant gut resistome.**
**a** Schematic timeline of antibiotic regimen. **b** Comparison of antimicrobial resistance genes (ARGs) NPS vs PS cohorts (overall $n = 53$ NPS, $n = 39$ PS). **c** Comparison of ARG classes count between NPS and PS (overall $n = 53$ NPS, $n = 39$ PS).
**d** Proportions of ARG types (NPS vs PS). The inset bar chart shows the mean ARG count. **e** Distribution of individual ARG profiles (NPS vs PS). **f** Venn diagram showing the number and percentage of unique and shared ARGs. **g** Comparison of Chao1 (genus richness) and Shannon (ecological diversity) indices between antibiotic-treated ($n = 27$ NPS, $n = 31$ PS) and control (untreated; $n = 26$ NPS, $n = 8$ PS) groups.
**h** Gut microbiome dynamics over the first three weeks of life, showing the top 10 genera. **i** Changes in taxonomic abundance (top 10 genera) in the antibiotic-treated versus control groups (NPS + PS). Shaded area indicates the antibiotic treatment period (median: 3 days). **j** Comparison of the number of ARGs between antibiotic-treated ($n = 27$ NPS, $n = 31$ PS) and control (untreated; $n = 26$ NPS, $n = 8$ PS) groups in both cohorts. **k** Comparison of the ARG count between NPS ($n = 27$ Antibiotic,

$n = 26$ Control) and PS ($n = 31$ Antibiotic, $n = 8$ Control) cohorts, stratified by antibiotic exposure. **l** Correlation plot of Shannon diversity index versus ARG number (NPS vs PS). τ represents Kendall's rank correlation coefficient. Shaded error bands indicate 95% confidence intervals around the regression lines. **m** Kendall's correlation between the number of ARGs and relative abundance of the top 10 most abundant genera. **n** Genomic mapping of an extracted contig containing the *mcr-9.1* gene against a public reference *mcr-9.1* gene from *E. coli*, showing 100% nucleotide identity. In (**b**, **c**, **g**, **j** and **k**), the box plots represent median (line inside the box), interquartile range (IQR; middle 50% of the data, box height), data within 1.5 × IQR (whiskers), and outliers (points). In (**b**, **c**, **d**, **g**, **i**, **j** and **k**), statistical significance was assessed using two-sided Wilcoxon tests with Benjamini–Hochberg correction. In (**l** and **m**), correlations were assessed using two-sided Kendall's rank correlation test. Statistical significance: *$P < 0.05$, **$P < 0.01$, ***$P < 0.001$, ****$P < 0.0001$; ns not significant.

patients (55-87% for very to extremely preterm infants in the UK)[37], allowing us to begin disentangling the specific effects of probiotics and antibiotics on the preterm gut microbiome and resistome. This carefully selected sub-cohort ($n = 34$) from a larger observational study ($n = 234$) consists exclusively of preterm infants (<33 weeks' gestation) who were matched for diet and birth mode, providing a relatively uniform baseline[8]. Moreover, in cases where empirical antibiotic therapy was administered, the duration and types of antibiotics (benzylpenicillin and/or gentamicin, the most frequently prescribed antibiotics in UK NICUs[37]) were strictly limited and standardised to less than 1 week. Despite antibiotic exposure, we observed minimal impact on overall microbiome diversity in both PS and NPS cohorts, suggesting that short-term early-life antibiotic treatment may not have an immediate or lasting effect on preterm microbiome diversity. In the PS cohort, daily probiotic supplementation likely facilitated rapid recovery of *Bifidobacterium*, while in the NPS cohort, the post-antibiotic increase in *Bifidobacterium* may reflect reduced competition or newly available ecological niches following antibiotic-mediated clearance of pathobionts. However, these patterns remain speculative, particularly given the short 3-week sampling window, and underscore the need for longer-term studies to fully understand how antibiotics, probiotics, and microbial community dynamics interact over time. This observation aligns with previous reports showing minimal immediate antibiotic effects but suggests that more pronounced shifts may emerge over extended periods[38].

However, the administration of antibiotics, though limited to an average duration of 3 median days, did specifically impact microbiome composition by promoting the proliferation of certain genera with pathogenic potential, including *Klebsiella* and *Enterococcus*, in the first week, with some of these strains persisting into the second week. *Enterobacter*, a genus less associated with multiple ARGs in this study, was observed to decline in the NPS cohort after antibiotic exposure, indicating potential susceptibility to the antibiotic regimen. In contrast, *Bifidobacterium* - particularly in the PS cohort - appeared to exert a protective effect, as evidenced by a marked reduction in *Enterococcus* abundance. This decrease could potentially be attributed to the carbon source depletion promoted by *Bifidobacterium* specialised HMO utilisation in human milk-fed gut environment, which may inhibit pathobiont colonisation and growth[39,40].

Overall, we identified a diverse range of ARGs in the gut microbiomes of preterm infants, with key ARG classes including genes encoding resistance to aminoglycosides, macrolides-lincosamides-streptogramins (MLS), beta-lactamases, trimethoprim, and tetracyclines. These ARG classes are clinically relevant in NICUs, where aminoglycosides and beta-lactams, e.g. gentamicin and benzylpenicillin, are frequently empirically administered from birth to cover possible infection[8,37]. The presence of these ARGs, underscores the potential for reduced antibiotic efficacy in this vulnerable population and highlights the need for careful and improved antibiotic

stewardship in NICU settings. Efflux pumps, which facilitate multidrug resistance by expelling a broad range of antibiotics[41], were also prevalent across multiple strains, particularly within *Escherichia* and *Klebsiella*, further complicating treatment options for infections by these pathogens in preterm infants. Notably, we detected a *mcr-9.1* gene, conferring resistance to colistin - a last-resort antibiotic -within the gut microbiome of a NPS infant. This gene was identical to a homolog found in *E. coli* strain 68A though it could not be definitively linked to a specific bacterial species in our cohort due to limitations in short-read sequencing data, as it appeared only as a single contig (Fig. 2m). The *mcr-9.1* gene was first identified in *Salmonella enterica* in 2019[42], where it conferred phenotypic resistance to colistin, and has since been associated with *Enterobacter* and *Klebsiella* species, often on mobile IncHI2 plasmids[43]. Our finding aligns with previous retrospective studies[43,44] that identified *mcr-9* prior to its first official report, highlighting the value of metagenomics in AMR surveillance. The presence of *mcr-9.1* in our cohort raises concerns about the silent circulation of colistin resistance genes within the gut microbiome of preterm infants, which may complicate future treatment options, particularly in cases where conventional antibiotics fail.

We also observed an increase in ARG abundance over time in NPS cohort, indicating that ARG acquisition may be driven by the introduction of new bacteria, likely hospital-acquired, rather than ARG evolution within the closed ecological system of the infant gut (at least across the short sampling widow). This is further supported by our strain-level analyses, where we observed the persistence of certain pathobionts, such as *Enterococcus*, across multiple time points. This persistence, along with the detection of shared strains among infants within the same hospital, suggests a high likelihood of nosocomial transmission of MDR bacteria. These findings align with reports from other NICU studies, where *Enterococcus*, *Escherichia*, and *Klebsiella* are common nosocomial pathogens and are often implicated in late-onset infections in preterm infants[32,45,46]. Indeed, we identified specific STs linked to neonatal infections, such as *E. coli* ST1193[47], ST73 and ST95[47,48], and *K. pneumoniae* ST432[49] strains, which are typically drug-resistant strains and of clinical significance. *Enterococcus*, frequently identified as a prominent ARG carrier in our cohort, exhibited multiple STs, including novel ones not previously reported, and certain strains were shared across different individual infants within the same hospital. This sharing among unrelated infants further supports the notion of nosocomial transmission of *Enterococcus*, a common cause of late-onset infections in NICUs. Thus, the frequent colonisation with hospital-acquired, ARG-bearing bacteria in the absence of probiotics highlights the importance of continuous surveillance and infection control in NICUs.

Our findings on the high prevalence of strains carrying plasmids, and in particular MDR plasmids, is particularly concerning, as it indicates that these strains could serve as reservoirs for resistance genes that may spread to other non-MDR strains within the same species as

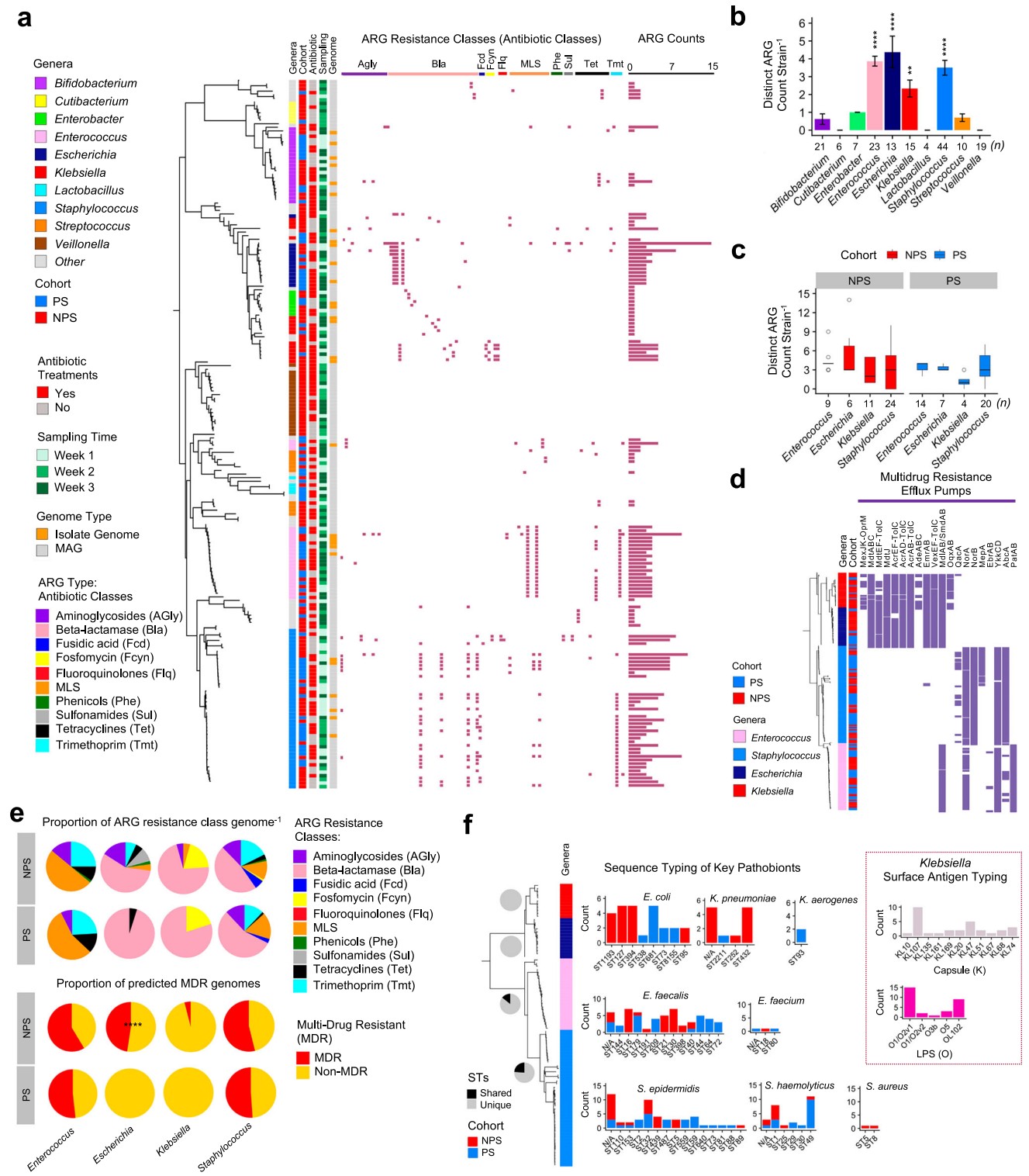

well as other genera via HGT. Notably, our findings indicate that antibiotic treatment promotes a higher prevalence of HGT events within the neonatal gut, likely due to the selective pressure antibiotics place on microbial populations. This pressure appears to favour the retention and spread of mobile genetic elements carrying ARGs, increasing HGT dynamics in antibiotic-treated infants. Indeed, our plasmid transfer experiment within *E. faecium* further emphasises the significance of plasmid-mediated ARG transfer in the neonatal gut as we demonstrated the successful transfer of a ~137 kb mega-plasmid carrying an aminoglycoside resistance gene (*aac6-aph2*) to a plasmid-free, gentamicin-sensitive recipient strain, conferring phenotypic

gentamicin resistance. It is important to note that our genomic analysis indicated only a weak correlation between overall ARG abundance and plasmid replicon count, suggesting that plasmids are not the sole contributors to ARG dissemination, and alternative routes of HGT are likely driving ARG spread[50].

The high-resolution sequencing data in this study underscored the beneficial impact of probiotic supplementation on microbiome development in preterm infants. PS infants showed an earlier and more robust colonisation of infant-associated *Bifidobacterium* species, particularly *B. breve* and *B. longum* subsp. *infantis*, compared to NPS infants[51–53]. The probiotic *B. bifidum* likely facilitated the establishment

**Fig. 3 | Strain-level resistome analysis. a** Neighbour-joining tree of 195 representative bacterial strains (post-genome dereplication) aligned with clinical metadata, ARG profiles (by resistance class), and ARG counts, the original dataset comprises both isolate genomes ($n = 89$) and metagenome-assembled genomes (MAGs; $n = 322$), totalling 411 individual bacterial genomes. **b** Comparison of distinct ARG counts per strain across the ten most abundant genera. Statistical analysis was performed using the Kruskal–Wallis test followed by Dunn's post hoc test (FDR-adjusted). Significance was compared against *Bifidobacterium*. \*\**P* < 0.01, \*\*\*\**P* < 0.0001. Data are presented as mean ± SEM. *n* represents number of representative strain-level genomes (total $n = 162$). **c** Comparison of ARG counts per strain between the four most resistant genera: *Enterococcus*, *Escherichia*, *Klebsiella* and *Staphylococcus*. Cohort comparisons were made using two-sided Wilcoxon tests with Benjamini–Hochberg correction, no difference were statistically significant. The box plot represents median (line inside the box), interquartile range (IQR; middle 50% of the data, box height), data within 1.5 × IQR (whiskers), and outliers (points). *n* represents number of representative strain-level genomes (total $n = 162$). **d** Profiles of multidrug resistance (MDR) efflux pumps ($n = 20$) across genomes ($n = 204$) of the four most resistant genera: *Enterococcus*, *Escherichia*, *Klebsiella* and *Staphylococcus*. **e** (Top) Proportions of ARG resistance classes per genome ($n = 204$) in both NPS and PS cohorts. (Bottom) Proportions of MDR versus non-MDR genomes in NPS and PS cohorts. MDR was defined as harbouring predicted ARGs conferring resistance to three or more antibiotic classes. Statistical significance was assessed using two-sided Fisher's exact test. \*\*\*\**P* < 0.0001. **f** Multilocus sequence typing (MLST) analysis of key pathobiont genomes ($n = 193$) from the four most resistant genera. Pie charts display genus-specific proportions of unique (non-shared) and shared sequence types between NPS and PS cohorts. Additional surface antigen typing (Kaptive) for *Klebsiella* spp. is shown in the inset box.

of these beneficial species through cross-feeding on HMOs[54,55], promoting a microbiome composition more similar to that of term infants. Indeed, this early and abundant presence of multiple *Bifidobacterium* species appeared to enhance pathogen colonisation resistance significantly, as evidenced by the limited presence of pathobionts, such as *Klebsiella* and *E. coli*, within the PS cohort.

Consistent with previous studies, our results also showed that PS infants harboured significantly fewer ARGs than NPS infants[29]. *Bifidobacterium*, and *Bifidobacterium* supplementation has been associated with lower ARG abundance in the infant gut microbiome, likely due to the competitive exclusion of ARG-rich pathobionts[56–59]. In our study, the higher abundance of *Bifidobacterium* in the PS cohort correlated with a reduced ARG load, particularly in genera commonly associated with MDR pathogens, such as *Enterococcus*, *Escherichia*, and *Klebsiella*. Notably, *Escherichia* and *Klebsiella* genomes in PS infants did not exhibit MDR characteristics, highlighting the potential for probiotic supplementation to mitigate MDR pathogen prevalence in the preterm gut. However, while plasmid analysis revealed a greater number of plasmids in the NPS cohort compared to PS infants, suggesting that probiotics may help reduce the plasmid pool carrying ARGs and, by extension, limit the ARG reservoir, we still observed plasmid transfer ex vivo even in the presence of high levels of *Bifidobacterium*, suggesting that presence of beneficial bacteria may not fully inhibit HGT. This observation underscores a critical point: while probiotics such as *Bifidobacterium* support beneficial microbiome composition and may suppress ARG-rich pathobionts, they do not necessarily prevent HGT or plasmid transfer, particularly in the antibiotic-exposed gut where HGT may be more common. This emphasises the need for further studies to evaluate the role of probiotics not only in microbial colonisation but also in their potential impact on HGT dynamics, particularly in environments where antibiotics are heavily used.

Finally, one limitation of our study was the relatively short sampling period, which cannot capture longer-term impacts of antibiotics and probiotics on microbiome development. Additionally, this study involved samples from two separate hospital sites, enabling comparison between baseline and probiotic-supplemented conditions. While this design may introduce potential confounders related to local environmental or clinical practice differences, our prior larger (parent) study of 234 infants found no significant site-specific effects on preterm microbiome profiles. Therefore, while we recognise this as a potential limitation, we consider any environmental influence on microbial composition and ARG profiles in the present study to be minimal and unlikely to alter our main conclusions. Nonetheless, we acknowledge the potential contribution of hospital-specific bacterial communities to the observed ARG profiles as evidenced by the high number of unique STs in each site. Future studies involving larger sample sizes, extended sampling windows, and multi-site designs could provide more comprehensive insights into the dynamic interactions between antibiotics, probiotics, and the neonatal microbiome. Furthermore, certain very low-abundance ARGs may have gone undetected due to limited sequencing coverage though rarefaction analysis showed sufficient coverage to capture species richness (Supplementary Fig. 5a). In our study, 7.9% of ARGs were identified in isolate genomes but not in the corresponding metagenomes (Supplementary Fig. 5b–h). Notably, all of these were singletons - each detected only once in the isolate genomes - highlighting that low-abundance ARGs can sometimes be missed in metagenomic analyses. This also underlines the importance of culturomics, and/or direct quantification methods (e.g. qPCR), as a complementary approach for capturing rare resistance genes. Additionally, gene-level rarefaction analysis may help determine the sequencing depth required for future studies of preterm infant stool samples.

In conclusion, our investigation provides a comprehensive overview of the preterm gut resistome and microbiome, demonstrating the role of probiotic supplementation in reducing ARG prevalence and pathogen load. However, the persistence of MDR pathogens such as *Enterococcus*, with its high plasmid carriage and demonstrated capacity for ARG transfer, even in a *Bifidobacterium*-dominant ecosystem, underscores the need for continued surveillance and targeted intervention strategies in NICUs to minimise the risk of colonisation and subsequent infections by MDR bacteria. Our findings highlight the complex interplay between antibiotics, probiotics, and HGT in shaping the neonatal microbiome, and provide a platform for further research into the role of probiotics in antimicrobial stewardship and infection control in vulnerable preterm populations.

## Methods
### Cohort and sample selection
A sub-set of 92 samples in total were selected from a previously published observational cohort study[8]. All NICUs in this study applied similar protocols for feeding, prescription of antibiotics and the use of prophylactic antifungal drugs. Hospital for PS cohort routinely prescribed all VLBW infants with oral probiotic supplementation (Infloran®, Desma Healthcare, Switzerland) twice daily (from birth until ~34 weeks post-menstrual age), while infants in NPS cohort were not supplemented with probiotics. Probiotic supplements contained *Bifidobacterium bifidum* ($1 \times 10^9$ cfu) and *Lactobacillus acidophilus* ($1 \times 10^9$ cfu). Samples from both cohorts (NPS and PS) were selected for this study based on the following criteria: VLBW preterm infants with gestational age <34 weeks, infants were solely given antibiotics benzylpenicillin and/or gentamicin, and age-matched infants without antibiotics, infants were all fed with breastmilk and/or donor breastmilk in both cohorts. A total of 34 VLBW preterm infants were selected for this study. Longitudinally collected faecal samples at week 1, week 2 and week 3 (first 3 weeks of their NICU stay) were recruited into this study (Supplementary Data 1).

### Cohort characteristics
This study is a 'controlled' sub-study from the original observational longitudinal study - Baby-Associated Microbiota of the Intestine

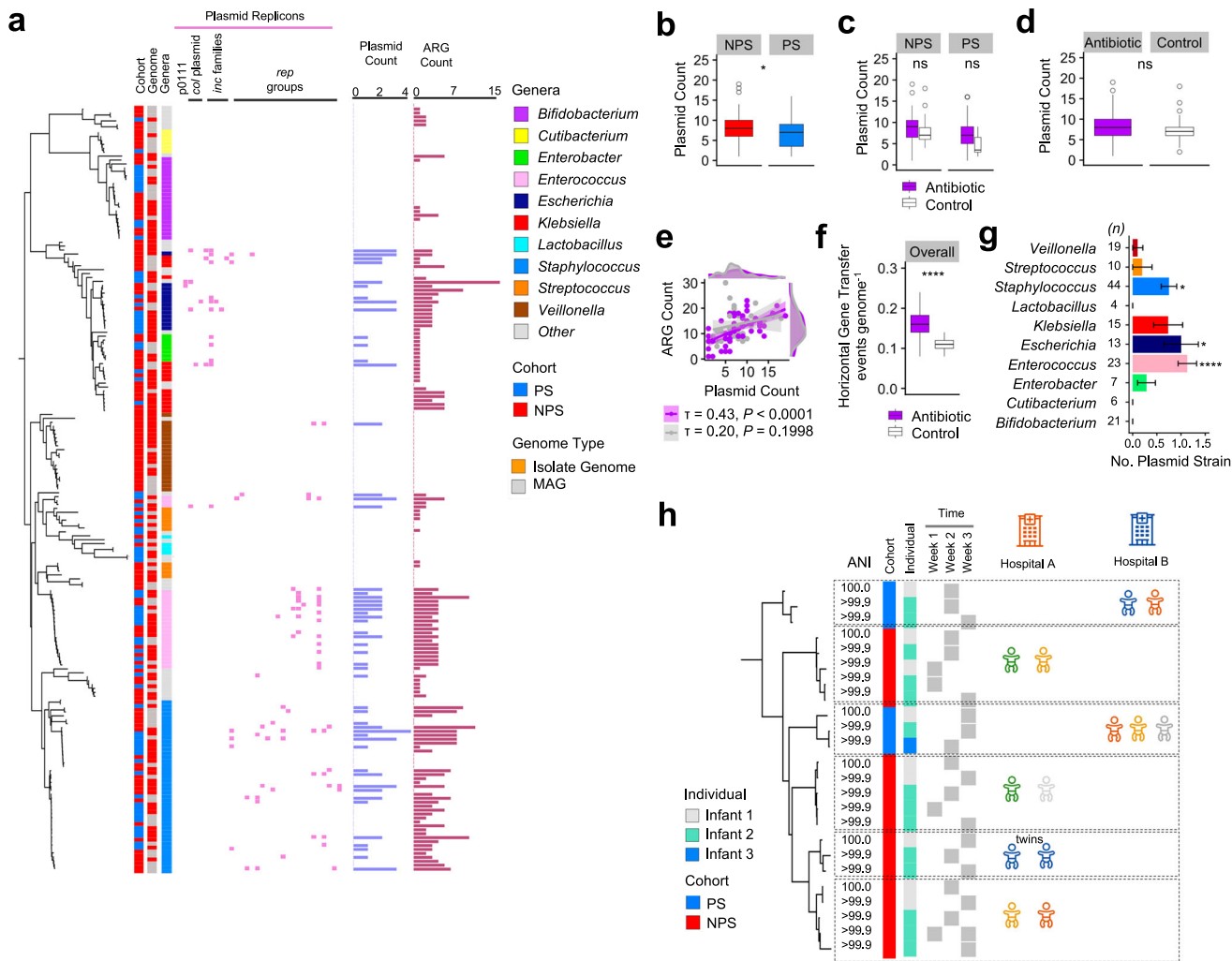

**Fig. 4 | Mobilome analysis and potential strain dissemination. a** Plasmid profiles of preterm infant gut-associated bacterial strains ($n = 195$) aligned with a strain-level neighbour-joining tree, cohort data, genome type, and bacterial taxonomy (highlighting the top 10 most abundant genera). Plasmid and ARG counts are also shown for comparison. **b** Comparison of gut microbiome plasmid counts between NPS ($n = 53$) and PS ($n = 39$) cohorts. *$P < 0.05$. **c** Comparison of plasmid counts between antibiotic-treated ($n = 27$ NPS, $n = 31$ PS) and control (untreated; $n = 26$ NPS, $n = 8$ PS) groups, stratified by cohort (NPS vs PS). *ns*, not significant. **d** Plasmid count comparison between antibiotic-treated ($n = 58$) and control ($n = 34$) groups across all preterm infant metagenome samples. ns not significant. **e** Correlation plot of ARG count versus plasmid count across all preterm infant metagenome samples. The scatter plot is coloured by antibiotic exposure: purple indicates antibiotic-treated infants; grey denotes untreated controls. $\tau$ represents Kendall's rank correlation coefficient. Shaded error bands indicate 95% confidence intervals around the regression lines. Correlation was assessed using a two-sided Kendall's rank correlation test. **f** Horizontal gene transfer (HGT) events per genome comparison

across all bacterial genomes ($n = 411$) between antibiotic-treated ($n = 237$) and control ($n = 174$) infants. ****$P < 0.0001$. **g** Comparison of plasmid replicon counts per strain across the ten most abundant genera. Statistical significance was assessed using the Kruskal–Wallis test followed by Dunn's post hoc test (FDR-adjusted). Significance was determined relative to *Bifidobacterium*. *$P < 0.05$, ****$P < 0.0001$. Data are presented as mean ± SEM. $n$ represents number of representative strain-level genomes ($n = 162$). **h** Potential dissemination or circulation of *Enterococcus* strains among preterm infants in NICUs at two separate hospital sites, with temporal information indicated. Each phylogenetic cluster represents clonal strains. Strain-level clustering defined by ANI > 99.9%. Each infant symbol represents an individual (singleton), unless otherwise indicated. In (**b**–**d** and **f**), the box plots represent median (line inside the box), interquartile range (IQR; middle 50% of the data, box height), data within 1.5 × IQR (whiskers), and outliers (points). In (**b**–**d** and **f**), statistical significance was assessed using two-sided Wilcoxon tests with Benjamini–Hochberg correction.

(BAMBI) study[8] as published previously. Summary of the cohort characteristics are presented in Supplementary Table 1. Faecal slurry for the plasmid transfer experiment was originally from five infants recruited in the PEARL study[60].

### Ethical approval
Faecal sample collection from Norfolk and Norwich University Hospital (BAMBI study) was approved by the Faculty of Medical and Health Sciences Ethics Committee at the University of East Anglia (UEA), and followed protocols laid out by the UEA Biorepository (License no: 11208). Faecal sample collection Imperial Healthcare NICUs (NeoM study) was approved by West London Research Ethics Committee

(REC) under the REC approval reference number 10/H0711/39. In all cases, medical doctors and nurses recruited infants after parents gave written consent. The PEARL study has been reviewed and agreed by the Human Research Governance Committee at the Quadram Institute Bioscience and the London-Dulwich Research Ethics Committee (reference 18/LO/1703) and received written ethical approval by the Human Research Authority. IRAS project ID number 241880[60].

### Genomic DNA extraction and Shotgun Metagenomic Sequencing
FastDNA Spin Kit for Soil (MP Biomedicals) was utilised to extract genomic DNA from infant faeces following manufacturer instructions,

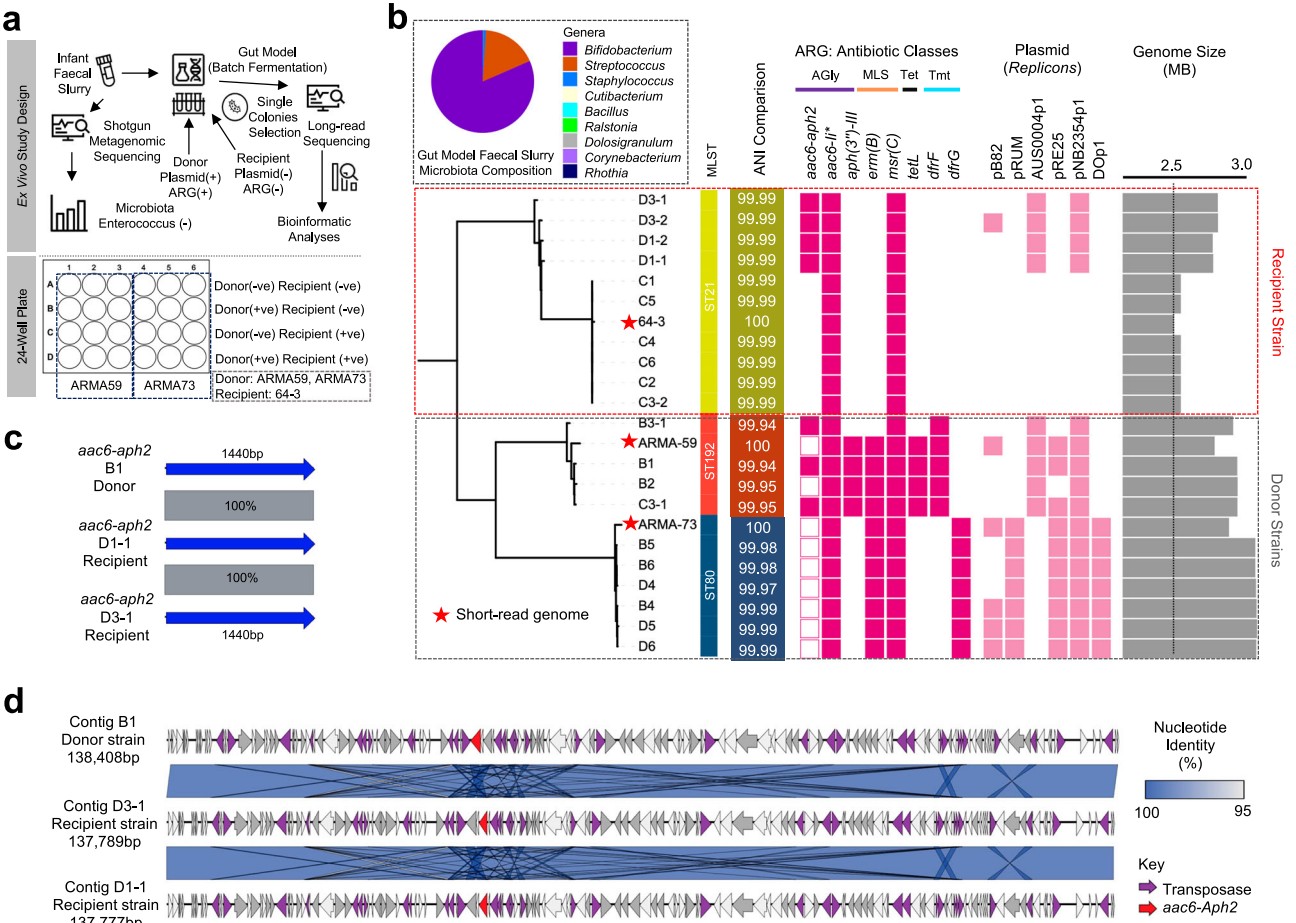

**Fig. 5 | Gentamicin-resistant plasmid transfer study in *Enterococcus*.**
**a** Schematic of the ex vivo study workflow and associated analyses, illustrating the study design within a 24-well plate format. **b** Antibiotic resistance gene profiling of donor and recipient strains. Colonies were individually picked from selective agar plates supplemented with the appropriate antibiotics and subjected to whole-genome sequencing using a combined long-read and short-read sequencing platform. A neighbour-joining tree was constructed to depict the relatedness between isolate genomes, aligned with plasmid replicons predicted within the genomes alongside their calculated genome sizes. The inset (top left) shows shotgun metagenomic contents (microbial taxa) of the infant faecal slurry utilised in the gut model experiment. Gene *aac6-Ii* is known to be a chromosomally encoded aminoglycoside acetyltransferase in *Enterococcus* spp. ARMA-59 and ARMA-73 are preterm-derived isolates (donor strains) whereas 64-3 (or, 64/3) is a plasmid-free laboratory strain. **c** Comparison of the computationally extracted antibiotic resistance gene *aac-aph2* from the representative donor strain B1 and recipient (transformant) strains D1-1 and D3-1. **d** Comparison of complete contig sequences encoding the gentamicin resistance gene *aac6-aph2*, predicted to be either whole or partial sequences of potential mobile genetic elements, including plasmids.

with an extended 3 min bead-beating on FastPrep tissue homogeniser (MP Biomedicals), additionally DNA was eluted with 55 °C sterile pure water. Genomic DNA concentration quantification followed using a Qubit 2.0 fluorometer (Invitrogen). DNA samples were then subject to established Illumina paired-end library preparation prior to Shotgun Metagenomic Sequencing on Illumina HiSeq 2500 to generate 125-bp paired-end reads (FASTQ), which was performed at the Wellcome Trust Sanger Institute (Supplementary Data 2).

**Bacterial isolation work and DNA extraction**
A targeted bacterial isolation work was carried out to isolate the most abundant taxa present in the preterm infant faecal samples – *Bifidobacterium*, *Enterococcus*, *Staphylococcus*, *Klebsiella* and *Escherichia* genera. Briefly, 25–50 mg of faecal samples were homogenised in 5 ml of phosphate buffer saline (PBS) by vortexing. Homogenates were then serially diluted to $10^{-4}$ in PBS and aliquots of 100 ml each were spread plated on selective agar respectively: MacConkey (Oxoid; targeting Gram-negative bacteria including *Escherichia* and *Klebsiella*), De Man-Rogosa-Sharpe (MRS; Difco) with 50 mg/L mupirocin (Oxoid; targeting *Bifidobacterium*), Baird-Parker (Oxoid; targeting *Staphylococcus*) and Slanetz and Bartley (Oxoid; targeting *Enterococcus*). Agar plates were

then incubated both aerobically (MacConkey, Baird-Parke, and Slanetz and Bartley) and anaerobically (MRS agar only) at 37 °C for 3 days. Next, 5 colonies from each agar plate were picked and were re-streaked for 3 consecutive times onto newly prepared agar plates to obtain pure isolates.

Bacterial isolates were cultured in appropriate media for overnight prior to genomic DNA extraction. DNA extraction was performed using the phenol-chloroform extraction method as described previously[61]. Briefly, PBS-washed bacterial cell pellets were resuspended in 2 ml of 25% sucrose in 10 mM Tris and 1 mM EDTA at pH 8.0, followed by enzymatic lysing step using 50 μl of lysozyme at 100 mg ml⁻¹ (Roche). Next, 100 μl of Proteinase K at 20 mg ml⁻¹ (Roche), 30 μl of RNase A at 10 mg ml⁻¹ (Roche), 400 μl of 0.5 M EDTA at pH 8.0 and 250 μl of 10% Sarkosyl NL30 (Thermo Fisher Scientific) were added sequentially into the lysed suspension. The suspension was then subjected to an 1 h incubation on ice and then a 50 °C water bath for overnight. On the next day, 3 rounds of phenol-chloroform-isoamyl alcohol (Merck) extraction were performed using 15 ml gel-lock tubes (QIAGEN), followed by chloroform-isoamyl alcohol (Merck) extraction prior to ethanol precipitation step and 70% ethanol cell wash for 10 min (twice). The DNA pellets were then air dried overnight,

re-suspended in 300 µl of sterile pure water and stored in −80 °C freezer prior to further analyses.

## Bacterial pure isolate Whole Genome Sequencing (WGS) and genome assembly

WGS of each pure bacterial isolate was performed on Illumina NextSeq 500 at the Quadram Institute (Norwich, UK) to generate 151-bp paired-end reads. Raw sequence reads (FASTQ) were firstly quality filtered (-q 20) with fastp v0.20.0[62] before de novo genome assembly via genome assembler optimiser Unicycler v0.4.9b[63] at default parameters that involved de novo genome assembler SPAdes v3.11.1[64], also Bowtie2 v2.3.4.1[65], SAMtools v1.7[66] and sequence polishing software Pilon v1.22[67]. Contigs with lengths <500 bp were discarded from each draft genome assembly before subsequent analysis. All draft genome assemblies underwent sequence contamination check via CheckM v1.1.3[68], and contaminated (sequence contamination >5%) and/or incomplete (genome completeness <90%) genome assemblies were excluded from further analysis ($n = 1$) resulting in a total of 89 high quality pure isolate genomes subject to subsequent genome analyses. Taxonomic assignment (species level identification) was performed using gtdb-tk v1.5.1[69]. Genome assembly statistics were generated via sequence-stats v1.0[70] and genome performed using Prokka v1.14.6[71] (Supplementary Data 3).

## Recovery of metagenome-assembled genomes

Shotgun metagenome raw reads (FASTQ) were firstly trimmed, adaptor-removed and quality-filtered using fastp v0.20.0[62] (-q 20). Subsequently, host-associated sequences were removed via Knead-Data v0.10.0 with human genome (GRCh38.p13 - no ALT) bowtie2 index file retrieved from https://benlangmead.github.io/aws-indexes/bowtie (with options --bypass-trim and --reorder) to generate clean FASTQ reads that consist of purely bacterial sequences. These metagenome reads were then co-assembled with MEGAHIT v1.2.9[72] prior to the reconstruction of Metagenome-Assembled Genomes (MAGs; Supplementary Fig. 6a, b). Next, MetaWRAP v1.3.2[73] was utilised to extract MAGs based on metagenome co-assemblies generated and metagenome clean reads via three binning software included in the pipeline MetaBAT v2.12.1[74], MaxBin v2.2.6[75] and CONCOCT v1.1.0[76] using sub-module *metawrap binning*. MAGs were then refined using sub-module *metawrap bin_refinement* to select the high-quality bins (MAGs) from each sample with completeness >90% and contamination <5% via checkm v1.1.3[68]. All MAGs were taxonomically ranked using gtdb-tk v1.5.1[69] via module *gtdbtk classify_wf*. A total of 411 high-quality (completeness >90%, contamination <5%) MAGs ($n = 322$) and isolate genomes ($n = 89$) were recovered for subsequent analyses (Supplementary Fig. 6c, d). Next, all MAGs and isolate genomes were dereplicated using dRep v3.2.2[77] at ANI 99.9% as the strain-level inference cut-off ($n = 195$).

## Taxonomic profiling for faecal metagenomes

After quality filtering steps as described above, briefly, shotgun metagenome raw reads (FASTQ) were quality-filtered using fastp v0.20.0[62] (-q 20) and host-associated sequences were removed using KneadData; Kraken v2.1.2[78] was utilised for taxonomic assignment for purified metagenome reads (Kraken2 standard Refseq indexes retrieved from https://benlangmead.github.io/aws-indexes/k2, May 2021), with confidence level set at 0.1. Bracken v2.6.2[79] was then deployed to re-estimate the relative abundance of taxa at both genus and species level (-t set at 10 as recommended to reduce false positive) from Kraken2 outputs as recommended. To simplify data for visualisation purpose, minimal genera with <1000 reads across all samples were removed prior to further analysis (with average 520,000 resultant genera reads across all samples). Minor genera with <2% (relative abundance) across all samples were classified as 'Others' to simplify representations prior to data visualisation in R[80] using library ggplot2[81].

## Phylogenetic relatedness estimation

The Mash-distance sequence tree consisted of 411 genomes (including MAGs) was generated using Mashtree v1.2.0[82] with 100 bootstrap replicates and option --mindepth 0, as in other cross-genus/species trees in this study. Distance tree was then mid-point rooted and visualised in iTOL v6[83].

## Metagenomic functional profiling

Filtered shotgun metagenomic raw reads as described in previous method section (using fastp v0.20.0) were concatenated into a single FASTQ file per sample prior to running Humann v3.0.0 and Metaphlan v3.0.13 based on CHOCOPhlAn_201901 metaphlan database (built using bowtie2) retrieved from https://zenodo.org/record/3957592#.YSZDVS1Q3_o. Path abundance output files (Supplementary Data 4) were then normalised and used for visualisation of microbiome functionalities (Supplementary Data 5).

## Antibiotic resistance genes, Multi-Locus Sequencing Typing (MLST) and plasmid replicons

Antibiotic resistance genes (ARGs) sequence search on all sequence data was performed via ABRicate v1.0.1 with options --minid = 95 and --mincov = 90 based on nucleotide sequence database ARG-ANNOT[84] NT v6 (retrieved from https://www.mediterranee-infection.com/acces-ressources/base-de-donnees/arg-annot-2/), individually validated against ResFinder v4.0[85] (only acquired ARGs) databases. The resistome analysis of infant gut metagenome was carried out via sequence search on metagenome co-assemblies of each sample, whilst strain-level resistome analysis was performed on individual genome assemblies. Antibiotic classes were sorted using outputs from ABRicate v1.0.1 based on antibiotic classes specified in ARG-ANNOT. Multidrug resistance efflux pumps were predicted using METABOLIC v4.0[86].

MLST was predicted via mlst v2.19.0[87] at default parameters through scanning contig files against the PubMLST[88] typing schemes (https://pubmlst.org/) sited at the University of Oxford.

Plasmid replicons were predicted using ABRicate v1.0.1[89] via PlasmidFinder sequence database v2[90] with options --minid = 90 and --mincov = 90 on individual strain-level genome assemblies. Plasmid replicons were reclassified in two key groups - *inc* families and *rep* groups, respectively prior to visualisation using iTOL v6[83].

## Klebsiella surface antigen typing

Kleborate v2.3.2[91] was invoked with the --all flag to type the capsule (K) and LPS (O) antigen loci, using default parameters.

## Horizontal gene transfer events analysis

Waafle v1.0[92] was used with default parameters to find incidences of horizontal gene transfer in the dereplicated MAGs and whole genome sequences. To determine statistical significance between the antibiotic positive ($n = 237$) and antibiotic negative ($n = 174$) cohorts, a random permutation test was conducted to normalise the sample sizes. Here, the number of HGT events predicted by Waafle from either population were randomly sub-sampled 100 times in GraphPad, and these values were then statistically tested using Wilcoxon test with Benjamini–Hochberg correction.

## *Enterococcus*-associated plasmid transfer study via ex vivo colon model

*E. faecium* ARMA59 and ARMA73 were isolated from preterm infant stool samples that were part of the BAMBI study. Isolates were selectively cultured from stool samples using Slanetz and Bartley Medium (Oxoid), on which *Enterococcus* forms dark red colonies. Colonies were then streaked on Brain Heart Infusion (BHI) agar (Merck) before being sent for whole genome sequencing. *E. faecium* 64/3, a plasmid-free derivative exhibiting high-level resistance to both rifampicin and fusidic acid, was utilized as the recipient strain for conjugation

assays. This strain was originally isolated from a stool sample of a hospital patient[93].

*E. faecium* ARMA59 and ARMA73 were used as donor strains in the ex vivo gut model experiment, while *E. faecium* 64/3 was used as a recipient strain. Both ARMA59 and ARMA73 were confirmed for their gentamicin resistance (>130 μg/ml) prior to the key experiment, while recipient strains 64/3 was phenotypically resistant to rifampicin (>25 μg/ml) and fusidic acid (>25 μg/ml).

Next, faecal slurry was prepared using *Enterococcus*-negative infant stool samples determined by selective agar plates (5 infants from the PEARL study). Stool samples (1965 mg) were mixed with 7 ml of reduced PBS to constitute the faecal slurry for the colon model experiment. Colon media were formulated based on our previous study with added vitamin solutions[94]. Briefly, 6 different media, M1 (500 ml), M2 (100 ml), M3 (100 ml), M4 (100 ml), M5 (200 ml), M6 (200 ml) were mixed aseptically in a sterile container. In addition, 200 ml of milk and vitamin mix (50 ml; pantothenate 10 mg/L, nicotinamide 5 mg/L, thiamine 4 mg/L, biotin 2 mg/L, vitamin B12 0.5 mg/L, menadione, 1 mg/L and p-aminobenzoic acid 5 mg/L) were also added, making the total volume 2000 ml, and this was adjusted to a maximum 700 ml final total volume (for each of the media or additives). These colon media were then pH-adjusted to pH6.8.

For bacterial strain preparation, frozen bacterial stocks were plated on BHI agar plates for 24 h anaerobically at 37 °C, followed by inoculating single colonies in 5 ml BHI broth for overnight (16 h) anaerobically at 37 °C. Next, the cultures were homogenised and mixed (200 μl) with 10 ml of colon media (equivalent to 1:50 dilution), incubated at 37 °C anaerobically overnight. The cultures were next washed (PBS + 3% cysteine) twice and pelleted.

Prior to experiments, faecal slurry was determined to be *Enterococcus*-free by both culturing (on Slanetz and Bartley Agar) and Whole Genome Sequencing approaches. DNA extraction of faecal slurry was performed via FastDNA Spin Kit for Soil (MP Biomedicals). Genomic DNA was then subjected to WGS via Illumina NovaSeq X to generate 151-bp paired-end reads. Raw sequencing reads (FASTQ) was processed as mentioned in previous section on metagenomic sequence analysis. Briefly, raw sequences were firstly quality-filtered with fastp v0.20.0 prior to taxonomic classification of sequence reads using Kraken v2.1.2[78] (Kraken2 standardDB; --confidence 0.1) and further classified by Bracken v2.8[79] (-t 10).

To set up the ex vivo colon model, the micro-Matrix fermentation system (Applikon Biotechnology) was used to model the human distal colon[95]. This system utilised a 24-well plate for batch fermentation. Four conditions were set up in the 24-well plate: 1) no donor, no recipient (control), 2) donor, no recipient, 3) no donor, recipient only, 4) donor and recipient strains. We have used a total volume of 5 ml in each well (slurry 26.6 μl, donor strain inoculum 100 μl, recipient strain inoculum 200 μl, colon media negative control 100/200 μl depending, colon media 4.67 ml). The basic parameters of the colon model was set at pH6.8, temperature 37 °C, and duration of experiment at 24 h. Samples were collected at 0 h and 24 h.

After the experiments, samples from 4 conditions then were re-streaked on Slanetz-Bartley agar plates supplemented or not with appropriate antibiotics: 1) no antibiotics, 2) 130 μg/ml gentamicin, 3) 25 μg/ml rifampicin + 25 μg/ml fusidic acid, 4) 130 μg/ml gentamicin + 25 μg/ml rifampicin +25 μg/ml fusidic acid, to allow the selection of donor and recipient strains, and transformants. After 48 h anaerobic incubation at 37 °C, pure colonies were selected and cultured in media broth to obtain bacterial pellets sufficient for DNA extraction. Genomic DNA extraction of each isolate was performed via FastDNA Spin Kit for Soil (MP Biomedicals) as per manufacturer's instructions. Well contents from 0 h were also sampled and DNA extracted with FastDNA Spin Kit for Soil (MP Biomedicals) for shotgun metagenome sequencing to confirm the absence of *Enterococcus*.

Genomic DNA was whole-genome sequenced on the Nanopore MinION platform (R10.4.1 flowcell) to generate long-read raw sequences with basecalling via Dorado[96]. Raw sequences were filtered using filtlong v0.2.1[97] to remove reads with <1000 bp. Filtered reads were then assembled via Flye v2.9[98] (--nano-hq −scaffold -g 3.6 m). Genome assemblies were subsequently polished using Medaka v1.11.3[99] (*medaka_consensus* -m r1041_e82_400bps_sup_v4.3.0) to generate high-quality genome assemblies for subsequent analyses.

Isolate DNA was also whole-genome sequenced on Illumina NextSeq 2000 to generate 151-bp paired-end short reads (FASTQ) in parallel with long-read sequencing as described in previous paragraph. Raw reads were firstly trimmed with fastp v0.20.0[62] (-q 20) prior to de novo genome assembly via genome assembler optimiser Unicycler v0.5.0[63] at default parameters to generate draft genome assemblies for further analyses.

### Average Nucleotide Identity (ANI) estimation and bacterial strain-level identification

ANI was computed using fastANI v1.34[100] at default parameters to compare genomes. Strain-level ANI cut-off was set at 99.9%, species-level cut-off was 95% (species boundary). To classify bacterial genomes at strain-level, dRep v3.2.2[27] (*dRep dereplicate --ignoreGenomeQuality*) was used at 99.9% ANI (-sa 0.999) to separate highly similar genomes.

### Estimation of bacterial replication rates, Average Genome Size and pathobiont sequencing coverage

Replication rates (Index of Replication) of *B. bifidum* were determined via iRep v1.1.7[101]. Briefly, *B. bifidum* Infloran genome was indexed using Bowtie v2.3.4.1[65] (--reorder), followed by mapping indexed files to metagenomic reads to generate alignment outputs (SAM file). Mapped reads were then used to estimate bacterial replication rates – by comparing (with genome assembly of *B. bifidum*) the sequencing coverage at near origin of replication and terminus. Index of Replication = 1 means no replication while >1 means active replication.

Average Genome Size (AGS) was computed via Microbe Census v1.1.0[102] at default parameters using host-free single concatenated FASTQ reads (from paired-end reads). Using the AGS estimation (based on universal single-copy genes in microbial genomes), the genome equivalents, the total coverage of microbial genomes in metagenome, were computed by dividing the total bases of metagenome with AGS. Genome equivalents were then used to normalise taxonomic abundance.

Bacterial DNA read coverage (depth) was estimated using CoverM v0.7.0[103]. To verify taxa abundance results, we assessed the read coverage of four key resistance-associated taxa: *E. faecalis*, *E. coli*, *K. pneumoniae*, and *S. epidermidis*. Reference genomes from pure isolates obtained in this study (NPS cohort) - *E. faecalis* N19-W3-G2, *E. coli* N19-W3-G3, *K. pneumoniae* N15-W3-G2, and *S. epidermidis* N10-W3-G1 - were used to estimate pathobiont coverage in the metagenomic samples. Briefly, metagenomic FASTQ paired-end reads were mapped to the reference genome assemblies using minimap2 v2.26[104] (default parameters), integrated within CoverM, which subsequently calculated coverage metrics including read depth, reads per base, and RPKM (Reads Per Kilobase per Million mapped reads).

### Data visualisation

Various bar plots, box plots, dot plots, line plots, pie charts and scatter plots were graphed via R v4.1.2[80], using R libraries tidyverse v1.3.1[105], ggplot2 v3.3.5[81] and ggpubr v0.6.0[106]. R library vegan v2.6.2[107] was used to construct non-metric multidimensional scaling (NMDS) plot and rarefaction curves (to a depth of 8 million reads) for gut microbiome (taxonomic/functional profiling) data. Genomic region of contigs/gene comparison was performed via R library genoplotr v0.8.11[108] or GenoFig v1.1.1[109]. Scatter plots with marginal histograms and regression

lines were generated using R library ggpubr v0.6.0[106] function *ggscatterhist*.

## Statistics

Statistical tests were performed via R base packages stats v4.1.2[80] or rstatix v0.6.0[110], including two-sided Wilcoxon's test (adjusted by Benjamini−Hochberg correction method where possible), Kruskal−Wallis test and Shapiro−Wilk normality test which was used to formally test for data normality where appropriate. Kendall's rank correlation tests were computed via R stats function *cor.test*. LEfSe[111] was used to perform linear discriminant analysis (LDA) on functional profiling output data from Humann3[112]. Shannon index, Chao1 index, absolute genus/species count of metagenome data, and PERMANOVA tests (function *adonis2*) were estimated via R library vegan v2.6.2[107]. Genome statistics were generated via sequence-stats v1.0[70]. R library dplyr v1.0.2[113] was used frequently for data handling.

## Reporting summary

Further information on research design is available in the Nature Portfolio Reporting Summary linked to this article.

## Data availability

Infant faecal sample metagenome sequencing raw reads generated in this study have been deposited in the NCBI Sequence Read Archive (SRA) under accession no. PRJNA1191223. Sequencing raw reads and draft genome assemblies for 89 isolates generated in this study have been deposited in the NCBI SRA and GenBank respectively, under accession no. PRJNA1191225. Sequencing raw reads and draft genome assemblies from long-read WGS in *Enterococcus* plasmid transfer study are available in SRA (for raw sequencing reads) and GenBank (genome assemblies) respectively under accession no. PRJNA1191226. Source data (also provided with this paper) and all 322 metagenome-assembled genomes recovered from gut metagenome samples are available via GitHub repository (https://github.com/raymondkiu/Infant-Resistome-Study) and Zenodo[114] (https://doi.org/10.5281/zenodo.15975760). Source data are provided with this paper.

## Code availability

No custom software was developed for this study. All analyses were performed using existing software and tools as described in the Methods.

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

## Acknowledgements

This research was supported in part by the Norwich Bioscience Institutes (NBI) Research Computing through the provision of a high performance computing cluster. We would like to thank the DNA sequencing teams at both Wellcome Trust Sanger Institute and Quadram Institute Bioscience for performing genomic DNA sequencing for this study. This work was

funded via Wellcome Trust Investigator Awar'sd to L.J.H. (100974/C/13/Z and 220540/Z/20/A) and support of the BBSRC Norwich Research Park Bioscience Doctoral Training Grant (BB/M011216/1; supervisor, L.J.H.; student, C.A.G.), Institute Strategic Programme (ISP) grant for Gut Microbes and Health BB/R012490/1 and its constituent project(s), BBS/E/F/000PR10353 and BBS/E/F/000PR10355 and a BBSRC ISP Food, Microbiome and Health BB/X011054/1 and its constituent project BBS/E/QU/230001B to L.J.H. W.v.S and L.J.H were also supported by BBSRC grant BB/S017941/1, and current funding by the National Institute for Health and Care Research (NIHR) Health Protection Research Unit in Public Health Genomics, a partnership between the UK Health Security Agency and the University of Birmingham. The views expressed are those of the author(s) and not necessarily those of the NIHR, the UK Health Security Agency or the Department of Health and Social Care. Work at Imperial Healthcare NICUs was supported by a programme grant from the Winnicott Foundation to J.S.K. and the National Institute for Health Research (NIHR) Biomedical Research Centre based at Imperial Healthcare NHS Trust and Imperial College London. K.S. was funded by an NIHR Doctoral Research Fellowship (NIHR-DRF-2011-04-128). We sincerely thank all clinical nurses at NNUH and Imperial Healthcare NICUs for collecting stool samples. We would like to give a special mention to research nurses Karen Few, Hayley Aylmer, and Zoe McClure for obtaining consent from parents and collecting samples.

## Author contributions

R.K., C.A. and L.J.H. conceived the study. R.K., A.A.G., L.E.L., S.P. and C.A. provided the methodology. R.K. and E.M.D. provided the software. L.J.H. and E.M.D. validated the study. R.K., E.M.D., A.A.G., L.E.L., A.C., S.P. and C.A. did the formal analysis. R.K., E.M.D., A.A.G., C.A. and A.C. carried out the investigations. K.S., L.E.L., A.G.S., P.C. and J.S.K. provided the resources. R.K., C.A. and K.S. curated the data. R.K. and L.J.H. wrote the original draft of the manuscript. W.vS., J.S.K. and L.J.H. supervised the study. R.K., W.vS., J.S.K. and L.J.H. reviewed and edited the manuscript. R.K. and C.A. administered the project. J.S.K. and L.J.H. acquired funds.

## Competing interests

The authors declare no competing interests.

## Ethics

This work complies with the inclusion and ethical guidelines upheld by Nature Communications.

## Additional information

¹Department of Microbes, Infection and Microbiomes, School of Infection, Inflammation and Immunology, College of Medicine and Health, University of Birmingham, Birmingham, UK. ²Institute of Microbiology and Infection, University of Birmingham, Birmingham, UK. ³Food, Microbiome and Health, Quadram Institute Bioscience, Norwich Research Park, Norwich, UK. ⁴Centro de Investigaciones en Microbiología y Biotecnología-UR (CIMBIUR), Facultad de Ciencias Naturales, Universidad del Rosario, Bogotá, Colombia. ⁵Health Sciences Faculty, Universidad de Boyacá, Tunja, Colombia. ⁶Faculty of Medicine, Imperial College London, London, UK. ⁷Norfolk and Norwich University Hospital, Norwich, UK. ⁸Norwich Medical School, University of East Anglia, Norwich, UK. ⁹These authors contributed equally: Elizabeth M. Darby, Cristina Alcon-Giner, Antia Acuna-Gonzalez. ✉e-mail: r.k.o.kiu@bham.ac.uk; l.hall.3@bham.ac.uk

