## [Transparent Peer Review file · Nature Communications]

Impact of early life antibiotic and probiotic treatment on gut microbiome and resistome of very-low-birth-weight preterm infants

Corresponding Author: Professor Lindsay Hall

Version 0:

Reviewer comments:

Reviewer #1

(Remarks to the Author)

The authors present a well-executed and interesting metagenomic study of fecal samples from two groups of preterm, very low birth weight (VLBW) infants: one group from a hospital where a probiotic supplement was administered (PS), and one group from a hospital that did not use probiotic supplementation (NPS). Within each group there were infants with and without antibiotic treatment. The cohort was curated to control for confounding factors such as diet and antibiotic course.

The authors claim that probiotic treatment increases the relative abundance of infant-associated *Bifidobacterium* species, and that antibiotic treatment increases the rate of horizontal gene transfer. The authors also claim that probiotic supplementation decreases the abundance and prevalence of antibiotic resistance genes (ARGs), pathogenic taxa, and plasmids. In surveying the metagenomes, they find a colistin resistance gene, several multidrug resistance efflux pumps, and several pathobionts, some of which appear to be nosocomial infections. The authors also demonstrate horizontal transfer of an antibiotic resistance plasmid between *Enterococcus* species in an ex-vivo study.

Major comments:

1. Figure 1 shows that the impact of PS on the VLBW infant microbiome is profound. It would be helpful to have a better understanding of the consistency of the patterns observed. Figures 1d-f show the relative mean abundance. How was this computed? In figure 1f, a violin plot, would help to assess whether the shift in the microbiome is observed consistently in the PS group, or dominated by outliers.
2. The authors do not observe major impact of antibiotic treatment on microbiome composition, which is interesting, but surprising. This observation should be caveated in the discussion section given the very low numbers of patients involved in some of the subgroups. This is understandable given the context of the study but should be mentioned.
3. Figure 1e shows that in the PS samples, *Bifidobacterium bifidum* accounts for roughly 50% of the measured species. Given that the probiotic supplementation includes a twice daily dose of 109 cfu *Bifidobacterium bifidum* throughout the course of the study, there is some question if this shows colonization and a durable shift in the microbiome. It would be helpful to add an analysis or rough estimate of how much the 109 cfu *B. bifidum* supplementation contributes to the sequencing results. Is *B. bifidum* replicating in the PS infants? The authors might also reference other literature on the durability of *B. bifidum* colonization after supplementation ends.
4. Beyond claiming decreased ARG abundance in PS, the authors also claim decreased ARG prevalence. This is hard to argue in sequencing data, since ARGs may be present at very low abundance but not detected because of dropouts in sequencing. If using the term "prevalence" this caveat should be addressed.
5. Almost all the infants were sampled across multiple timepoints, yet there is no analysis of trends across time using paired timepoint samples.

Minor comments:

1. In Fig. 1 the subfigure labeling in the caption is off starting with b and ending with f.
2. It is challenging to read the taxon labels in Fig. 1c.
3. Why is the value legend in square root of copies per million in Fig. 1i? This seems an unusual rescaling.
4. In the caption for Fig. 2e “bot” should be “both”
5. In Fig. 2l and Fig. 4e there should be statistics for both regression lines, not just for the overall rank correlation. Also, are the slopes of the two lines different?
6. In Fig. 4h how is it possible that an individual was assigned to both PS and NPS cohorts?

Reviewer #2

(Remarks to the Author)

Kiu and colleagues present an interesting study investigating the impact of probiotics and antibiotics on the gut microbiome and resistome of preterm infants. The use of shotgun metagenomics combined with culturing and ex vivo experiments provides valuable insights into microbial composition, antibiotic resistance gene (ARG) prevalence, and horizontal gene transfer (HGT) potential. Their findings contribute to the growing evidence supporting probiotic supplementation as a potential tool for antimicrobial stewardship in neonatal care.

While this work is significant, several methodological issues need to be addressed to strengthen the conclusions drawn. In particular, the study design introduces confounding factors that complicate the interpretation of probiotic effects. Additionally, concerns regarding compositional data biases and sequencing depth, in particular the potential dilution effect due to probiotics warrant further attention preferably by some level of absolute quantification. Below, I outline key areas that should be considered in revision.

Methodological issues related to the study design

- A limitation is that the probiotic-supplemented (PS) and non-supplemented (NPS) groups originate from different hospitals. This introduces potential environmental and institutional differences that could influence microbiome composition and ARG profiles beyond the effect of probiotics. While it is briefly addressed in the discussion, the authors should more explicitly acknowledge this and discuss how it may impact their conclusions. In addition, some exploratory analyses could be done to examine to what extent environmental difference have an effect in addition to probiotic influences (see section multi-locus sequence typing below).

I would suggest focussing less on the (lack of) statistical differences in baseline comparisons between the two cohorts and rather focus on apparent (non-statistically significant) differences between both cohorts for example in terms of gestational age and mode of delivery that are now not addressed. Given the observational nature of the study, baseline characteristic testing can be problematic as also indicated by the STROBE guidelines. Statistical tests of background or baseline characteristics can mislead the reader since they do not provide guidance about the magnitude of the confounding that could occur due to group imbalances or because the study is not powered for such comparisons. P-values for such tests should therefore not be presented or overinterpreted.

- The differential impact of antibiotics within the NPS cohort is particularly interesting (e.g., increased bifidobacteria abundance in antibiotic-treated infants), but this finding is underexplored in the discussion.

Compositional Data Bias and Need for Quantitative Validation

- The study relies on compositional sequencing data, which may overstate probiotic effects due to the dilution of other taxa in a low-biomass environment. Adding probiotics in such an ecosystem could artificially reduce the apparent abundance of ARGs and pathobionts. As depicted in Fig 1d approximately bifidobacteria contribute around 75% of the relative abundance data in the probiotic group which could result in substantial dilution effect. To confirm whether these decreases in the other microbial taxa are (in part) real, the authors should complement their data with qPCR for absolute quantification. This could either be quantitative microbiome profiling or absolute quantification of specific pathobionts.

- o The observed decrease in enterococci and MDR pathogens could be due to a coverage issue rather than true displacement. If the sequencing depth is similar across samples, but bifidobacteria dominate in the PS group, the detection limit for ARGs and other species will be lower. Quantifying absolute abundances of key taxa and ARGs would address this concern.

- o The lower microbial diversity in the PS group is likely due to the high proportion of supplemented bifidobacteria. However, it remains unclear whether this represents actual displacement of other species or a mere shift in relative abundance. The authors should also investigate microbial richness next to diversity and assess whether other bacterial taxa decrease in absolute terms.

- o The authors should consider providing read coverage statistics for pathobionts in PS vs. NPS datasets to confirm that the observed reduction is not a sequencing bias.

ARG Detection and Biases

- The differences in ARG profiles could be influenced by hospital-specific bacterial communities rather than probiotics alone. The authors should acknowledge that circulating hospital microbiomes might contribute to the observed ARG trends.
- The presence of *mcr-9* in samples collected from 2011–2012 is interesting but similar observations were also made for other *mcr* variants. Retrospective analyses both *in silico* on deposited metagenomes as well as on biobanked fecal samples or bacterial isolates have identified *mcr* genes in samples collected well before their first official reports. It could be stated that the current finding fits in a line of previous observations and highlights the value of the microbiome for AMR surveillance.
- It remains unclear to the reviewer whether MDR characteristics were assessed only from metagenome-assembled genomes (MAGs) or if cultured isolates from the PS and NPS groups also showed differences. The latter would provide stronger evidence that MDR pathogens were actually reduced rather than simply undetected due to lower sequencing coverage.

Multi-locus sequence typing

- The focus on *Enterococcus* HGT potential is valuable, but other species, but it would be interesting if the authors could also check whether the dominant sequence types (STs) of the other pathobionts differ between PS and NPS groups, as this could be indicative for environmental influences rather than probiotic-driven changes.

Statistical Considerations and Figure Clarifications

- Multiple comparison corrections should be applied in differential abundance analyses (e.g., Figure 1g).
- In Figure 1i, it would be helpful to specify which bacteria contribute to the unique differential pathways. Do these pathways primarily belong to bifidobacteria?
- Figure 2b should clarify what the number of ARGs represents (e.g., absolute counts, relative abundance?). Since higher bacterial diversity can naturally lead to more detected genes, including ARGs (Figure 2l), the authors might want to adjust for this in their analysis.

Minor Comments

- Line 108 contains an unnecessary large space.

Reviewer #3

(Remarks to the Author)

Version 1:

Reviewer comments:

Reviewer #1

(Remarks to the Author)

The authors have satisfactorily addressed the comments regarding consistency of abundance measurements (1), caveats on the surprisingly minor impact of antibiotic treatment (2), and the presentation of time series data (5). For the comment on colonization of *Bifidobacterium* (3), I recognize that quantification is not possible in this case, and the evidence the authors present addresses the comment.

The authors have also added appropriate caveats on ARG detection in the metagenomic data (comment 4). However, while the species-level rarefaction analysis is a convincing argument that taxonomic detection is not a problem, I suggest that a gene- or mechanism- level rarefaction analysis would be more relevant to the ARG detection discussion (e.g. Zaheer et al. *Sci Rep.* 2018. <https://doi.org/10.1038/s41598-018-24280-8>). Further, is it the contribution of very low abundance ARGs that explains why the decreased ARG abundance observed in the PS metagenomes (Fig. 2b) is not observed in the PS isolates (Supplementary Fig. 5b,c)? Last, Figure 4h, which presents interesting and important information, remains visually confusing to decipher because of difficulty mapping the infant symbols on the right to rows in the “cohort”, “individual”, and “time” columns. If I’m reading the plot correctly, perhaps extending the boxes separating values in the “ANI” column is a solution.

Reviewer #2

(Remarks to the Author)

The authors have done extensive revisions, including additional analyses (e.g. on replication rate of *B. bifidum*), nuancing the discussion and highlighting the consistency with prior findings (e.g., *mcr-9.1*) and clarification and further elaboration (e.g., the likely limited impact of study site as major confounding factor). While it remains a limitation that quantitative approaches can no longer be conducted due to lack of remaining samples/DNA, the authors have at least excluded the role

of mere dilution effects or sequencing depth bias as good as possible using computational approaches. All my comments and concerns have thereby been adequately addressed.

Reviewer #3

(Remarks to the Author)

REVIEWER COMMENTS

Reviewer #1 (Remarks to the Author):

The authors present a well-executed and interesting metagenomic study of fecal samples from two groups of preterm, very low birth weight (VLBW) infants: one group from a hospital where a probiotic supplement was administered (PS), and one group from a hospital that did not use probiotic supplementation (NPS). Within each group there were infants with and without antibiotic treatment. The cohort was curated to control for confounding factors such as diet and antibiotic course.

The authors claim that probiotic treatment increases the relative abundance of infant-associated *Bifidobacterium* species, and that antibiotic treatment increases the rate of horizontal gene transfer. The authors also claim that probiotic supplementation decreases the abundance and prevalence of antibiotic resistance genes (ARGs), pathogenic taxa, and plasmids. In surveying the metagenomes, they find a colistin resistance gene, several multidrug resistance efflux pumps, and several pathobionts, some of which appear to be nosocomial infections. The authors also demonstrate horizontal transfer of an antibiotic resistance plasmid between *Enterococcus* species in an ex-vivo study.

Major comments:

1. Figure 1 shows that the impact of PS on the VLBW infant microbiome is profound. It would be helpful to have a better understanding of the consistency of the patterns observed. Figures 1d-f show the relative mean abundance. How was this computed? In figure 1f, a violin plot, would help to assess whether the shift in the microbiome is observed consistently in the PS group, or dominated by outliers.

Thank you for this helpful comment. In the revised manuscript, Fig. 1d–e shows the mean relative abundance, calculated as the average of relative abundance across all samples within each cohort. We have updated the original Fig. 1f (now Fig. 1g) by replacing the bar chart with a box plot rather than a violin plot, as the sample size is too small for violin plot visualisation. The box plot allows us to display both the distribution and any outliers, and it clearly illustrates the consistent shift in *Bifidobacterium* species relative abundance within the PS group. We believe this addresses the reviewer’s point regarding consistency across samples.

2. The authors do not observe major impact of antibiotic treatment on microbiome composition, which is interesting, but surprising. This observation should be caveated in the discussion section given the very low numbers of patients involved in some of the subgroups. This is understandable given the context of the study but should be mentioned.

We have updated the discussion section by adding the following sentence (lines 285-295): “Despite antibiotic exposure, we observed minimal impact on overall microbiome diversity in both PS and NPS cohorts, suggesting that short-term early-life antibiotic treatment may not have an immediate or lasting effect on preterm microbiome diversity. In the PS cohort, daily probiotic supplementation likely facilitated rapid recovery of *Bifidobacterium*, while in the NPS cohort, the post-antibiotic increase in *Bifidobacterium* may reflect reduced competition or newly available ecological niches following antibiotic-mediated clearance of pathobionts. However, these patterns remain speculative, particularly given the short 3-week sampling

window, and underscore the need for longer-term studies to fully understand how antibiotics, probiotics, and microbial community dynamics interact over time. This observation aligns with previous reports showing minimal immediate antibiotic effects but suggests that more pronounced shifts may emerge over extended periods”. This explicitly acknowledges the caveat regarding subgroup size and the need for caution when interpreting the antibiotic-related findings.

3. Figure 1e shows that in the PS samples, *Bifidobacterium bifidum* accounts for roughly 50% of the measured species. Given that the probiotic supplementation includes a twice daily dose of 10⁹ cfu *Bifidobacterium bifidum* throughout the course of the study, there is some question if this shows colonization and a durable shift in the microbiome. It would be helpful to add an analysis or rough estimate of how much the 10⁹ cfu *B. bifidum* supplementation contributes to the sequencing results. Is *B. bifidum* replicating in the PS infants? The authors might also reference other literature on the durability of *B. bifidum* colonization after supplementation ends.

We thank the reviewer for making this important point. As this work builds on a subset of a previously published study, we have already demonstrated that the probiotic *Bifidobacterium bifidum* (but not the *Lactobacillus* component of Infloran) showed lasting colonisation, persisting >6 months after supplementation ended¹. We also isolated the *B. bifidum* Infloran strain from older infants in the study and thus confirmed replication.

In the revised manuscript, we have now added an additional computational analysis using iRep² to estimate bacterial replication rates of *B. bifidum* Infloran strains in the infant metagenomes. The results indicate active replication (iRep >1.5) of *B. bifidum* Infloran strain in the gut of all PS infants during their first 3 weeks of life (Fig. 1f), further supporting our previous lab findings and suggesting a durable shift in the microbiome. This is now added in results section lines 113-117.

We were unable to directly quantify how much the twice-daily 10⁹ CFU *B. bifidum* dose contributed to sequencing reads, as no additional sample material was available for direct measurement. However, several aspects support the interpretation that the observed *B. bifidum* sequences reflect true colonisation: (i) near-zero abundance of *B. bifidum* in NPS infants (0%; Fig. 1g), despite literature reports that this species can appear early in the infant gut; (ii) the detection of active replication by iRep; and (iii) the low abundance of *Lactobacillus* in PS infants, despite being present in the supplement, consistent with its known poor colonisation ability. Together, these data suggest that the probiotic *B. bifidum* is not merely transient but establishes itself within the PS infant gut.

4. Beyond claiming decreased ARG abundance in PS, the authors also claim decreased ARG prevalence. This is hard to argue in sequencing data, since ARGs may be present at very low abundance but not detected because of dropouts in sequencing. If using the term “prevalence” this caveat should be addressed.

We have added sentences to address this caveat in the discussion (Lines 409-417):
“Furthermore, certain very low-abundance ARGs may have gone undetected due to limited sequencing coverage though rarefaction analysis showed sufficient coverage to capture species richness (Supplementary Fig. 5a). In our study, 7.9% of ARGs were identified in isolate genomes but not in the corresponding metagenomes (Supplementary Fig. 5b-h). Notably, all of these were singletons - each detected only once in the isolate genomes -

highlighting that low-abundance ARGs can sometimes be missed in metagenomic analyses. This also underlines the importance of culturomics, and/or direct quantification methods (e.g. qPCR), as a complementary approach for capturing rare resistance genes.”

5. Almost all the infants were sampled across multiple timepoints, yet there is no analysis of trends across time using paired timepoint samples.

We have added an additional longitudinal analysis and supplementary figure based on antibiotic-exposed infants (Supplementary Fig. 3) to understand the trends of taxa changes. A sentence has been added (lines 166-169) to reflect this: “A further intra-cohort longitudinal analysis of antibiotic-exposed infants in both cohorts revealed an increase in *Bifidobacterium* abundance in the PS cohort by week 2 (compared to week 1), while overall *Staphylococcus* levels decreased across both cohorts in week 2 (Supplementary Fig. 3)”.

Minor comments:

1. In Fig. 1 the subfigure labeling in the caption is off starting with b and ending with f. We have checked all the figure captions and revised them.

2. It is challenging to read the taxon labels in Fig. 1c. We have redone Fig. 1c and made the taxon labels (driving taxa) more visible.

3. Why is the value legend in square root of copies per million in Fig. 1i? This seems an unusual rescaling. Copies per million values were obtained from the HUMAnN outputs; to facilitate visualisation (due to data skewness), these values were transformed for plotting purposes. This transformation does not affect the statistical outcomes or the interpretation of the data in any way.

4. In the caption for Fig. 2e “bot” should be “both”
Now updated.

5. In Fig. 2l and Fig. 4e there should be statistics for both regression lines, not just for the overall rank correlation. Also, are the slopes of the two lines different? We have now added the statistical details (Kendall coefficients and p-values) for both regression lines; although the slopes differ slightly, these values and their interpretations have been incorporated into the revised results section (lines 240-241).

6. In Fig. 4h how is it possible that an individual was assigned to both PS and NPS cohorts? No, it is not possible. Each infant symbol in Fig. 4h represents a unique individual (colour-coded). Hospital A belongs to the NPS cohort, while Hospital B belongs to the PS cohort. We have improved the figure legend in the revised manuscript to clarify this information.

Reviewer #2 (Remarks to the Author):

Kiu and colleagues present an interesting study investigating the impact of probiotics and antibiotics on the gut microbiome and resistome of preterm infants. The use of shotgun metagenomics combined with culturing and ex vivo experiments provides valuable insights into microbial composition, antibiotic resistance gene (ARG) prevalence, and horizontal gene transfer (HGT) potential. Their findings contribute to the growing evidence supporting probiotic supplementation as a potential tool for antimicrobial stewardship in neonatal care.

While this work is significant, several methodological issues need to be addressed to strengthen the conclusions drawn. In particular, the study design introduces confounding factors that complicate the interpretation of probiotic effects. Additionally, concerns regarding compositional data biases and sequencing depth, in particular the potential dilution effect due to probiotics warrant further attention preferably by some level of absolute quantification. Below, I outline key areas that should be considered in revision.

Methodological issues related to the study design

- A limitation is that the probiotic-supplemented (PS) and non-supplemented (NPS) groups originate from different hospitals. This introduces potential environmental and institutional differences that could influence microbiome composition and ARG profiles beyond the effect of probiotics. While it is briefly addressed in the discussion, the authors should more explicitly acknowledge this and discuss how it may impact their conclusions. In addition, some exploratory analyses could be done to examine to what extent environmental difference have an effect in addition to probiotic influences (see section multi-locus sequence typing below).

A PERMANOVA multivariate analysis (Fig. 1c) demonstrated that differences in microbial composition among infants in this study were significantly associated only with cohort (NPS vs. PS), and not with gestational age, antibiotic exposure, or birthweight - likely driven by the presence of *Bifidobacterium*. Importantly, this cohort is a subset of a larger published dataset (n = 234), published in 2020¹, in which we performed detailed multivariate and multilevel pairwise analyses. That prior work showed no significant differences in microbiome composition across three control hospital sites located in different geographical regions. Consistent with our current findings, that study demonstrated that probiotic supplementation, particularly the introduction of *Bifidobacterium*, was the main driver of microbiome compositional differences in supplemented versus non-supplemented preterm infants. We have updated the discussion to explicitly acknowledge this point as a limitation and explain why we believe its impact is minimal:

Lines 398-406: “Additionally, this study involved samples from two separate hospital sites, enabling comparison between baseline and probiotic-supplemented conditions. While this design may introduce potential confounders related to local environmental or clinical practice differences, our prior larger (parent) study of 234 infants found no significant site-specific effects on preterm microbiome profiles. Therefore, while we recognise this as a potential limitation, we consider any environmental influence on microbial composition and ARG profiles in the present study to be minimal and unlikely to alter our main conclusions”.

I would suggest focussing less on the (lack of) statistical differences in baseline comparisons between the two cohorts and rather focus on apparent (non-statistically significant) differences between both cohorts for example in terms of gestational age and mode of delivery that are now not addressed. Given the observational nature of the study, baseline characteristic testing can be problematic as also indicated by the STROBE guidelines.

Statistical tests of background or baseline characteristics can mislead the reader since they do not provide guidance about the magnitude of the confounding that could occur due to group imbalances or because the study is not powered for such comparisons. P-values for such tests should therefore not be presented or overinterpreted.

Thank you for this suggestion. We fully acknowledge that, as per the STROBE guidelines, statistical tests on baseline characteristics in observational studies can be misleading and are often discouraged, as they do not inform on the magnitude or impact of potential confounding and the study is not powered for these comparisons. In line with this recommendation, we have now removed the P-values from the cohort characteristics table in the revised manuscript to avoid over-interpretation. As noted above, our multivariate PERMANOVA analyses show that only the cohort variable (NPS vs. PS) significantly contributes to differences in microbiome profiles, while other factors - including gestational age and mode of delivery - do not significantly explain the observed variation.

- The differential impact of antibiotics within the NPS cohort is particularly interesting (e.g., increased bifidobacteria abundance in antibiotic-treated infants), but this finding is underexplored in the discussion.

Thank you for highlighting this point, and we agree that the differential impact of antibiotics within the NPS cohort, warrants further discussion. As shown in Fig. 2i, while taxonomic abundance analysis revealed no significant overall differences between antibiotic-exposed and unexposed NPS infants, *Bifidobacterium* (and to a lesser extent *Lactobacillus*) showed clear temporal dynamics: in week 1, *Bifidobacterium* abundance decreased due to antibiotic treatment (which on average lasted ~3 days), but in the exclusively human milk-fed infants, *Bifidobacterium* levels rebounded and even increased in weeks 2 and 3. This pattern is consistent with expectations based on ecological recovery after short-term antibiotic perturbation.

We have now expanded the discussion (lines 285-295) to better reflect these findings: “Despite antibiotic exposure, we observed minimal impact on overall microbiome diversity in both PS and NPS cohorts, suggesting that short-term early-life antibiotic treatment may not have an immediate or lasting effect on preterm microbiome diversity. In the PS cohort, daily probiotic supplementation likely facilitated rapid recovery of *Bifidobacterium*, while in the NPS cohort, the post-antibiotic increase in *Bifidobacterium* may reflect reduced competition or newly available ecological niches following antibiotic-mediated clearance of pathobionts. However, these patterns remain speculative, particularly given the short 3-week sampling window, and underscore the need for longer-term studies to fully understand how antibiotics, probiotics, and microbial community dynamics interact over time. This observation aligns with previous reports showing minimal immediate antibiotic effects but suggests that more pronounced shifts may emerge over extended periods”.

Compositional Data Bias and Need for Quantitative Validation

- The study relies on compositional sequencing data, which may overstate probiotic effects due to the dilution of other taxa in a low-biomass environment. Adding probiotics in such an ecosystem could artificially reduce the apparent abundance of ARGs and pathobionts. As depicted in Fig 1d approximately bifidobacteria contribute around 75% of the relative abundance data in the probiotic group which could result in substantial dilution effect. To confirm whether these decreases in the other microbial taxa are (in part) real, the authors should complement their data with qPCR for absolute quantification. This could either be quantitative microbiome profiling or absolute quantification of specific pathobionts.

Unfortunately, we no longer have access to the original samples for qPCR quantification, and the limited archived DNA we hold - stored at -20°C for over six years - is unlikely to yield reliable results. To address this limitation, we performed rarefaction analysis, which confirmed that sequencing depth was sufficient to capture the species diversity and richness of the relatively low-diversity preterm gut microbiota (Supplementary Fig. 5a). This was further supported by absolute genus/species richness measures and the Chao1 index, both showing close agreement between observed and estimated species counts.

To minimise the impact of unequal sequencing depth, we additionally calculated the Average Genome Size for each sample using MicrobeCensus and normalised the taxonomic profiles accordingly (Supplementary Fig. 2). The resulting data - expressed as read counts per genome equivalent - were consistent with relative abundance values and statistical significance (Fig. 1h), suggesting that sequencing depth did not bias the taxonomic composition. We added a sentence in lines 124-125 to reflect the evidence: “This was supported by read coverage data, which showed a consistent reduction in *Klebsiella* and *Escherichia* (Supplementary Fig. 2c-d).” Furthermore, we estimated *B. bifidum* Infloran strain (probiotic) replication rates (>1.5) from the metagenomic data, supporting that the detected sequences originated from actively replicating populations rather than residual DNA from supplementation. We also modified the sentence in lines 112-117 to reflect this outcome: “In contrast, the gut microbiomes of PS infants were dominated by the genus *Bifidobacterium*, particularly *Bifidobacterium bifidum*, a major component of the Infloran probiotic provided to the infants, exhibited active replication in the preterm gut, highlighting the impact of probiotic supplementation (Fig. 1f).” Together, these analyses provide strong evidence that the observed microbiome shifts reflect genuine biological changes rather than sequencing artefacts or dilution effects.

o The observed decrease in enterococci and MDR pathogens could be due to a coverage issue rather than true displacement. If the sequencing depth is similar across samples, but bifidobacteria dominate in the PS group, the detection limit for ARGs and other species will be lower. Quantifying absolute abundances of key taxa and ARGs would address this concern.

Please see previous comments. After normalising the relative abundance using genome equivalents, we observed similar trends and outcomes (Supplementary Fig. 2a–b). Sequencing coverage of key multidrug-resistant (MDR) pathogens also demonstrated a similar decreasing trend, supporting the relative abundance data we have (Supplementary Fig. 2c–d). We further computed the average ARG count in pure isolate genomes ($n = 89$), which showed comparable results to those of the metagenomes. Notably, 7.9% ($n = 9$) of ARGs were detected exclusively in isolate genomes, and not in the metagenomes or MAGs, indicating that rare resistance genes may sometimes be missed. However, the proportion is low, and such genes can be recovered through culturing approaches. This has now been addressed in the limitations section (lines 409-417): “Furthermore, certain very low-abundance ARGs may have gone undetected due to limited sequencing coverage though rarefaction analysis showed sufficient coverage to capture species richness (Supplementary Fig. 5a). In our study, 7.9% of ARGs were identified in isolate genomes but not in the corresponding metagenomes (Supplementary Fig. 5b-h). Notably, all of these were singletons - each detected only once in the isolate genomes - highlighting that low-abundance ARGs can sometimes be missed in metagenomic analyses. This also underlines the importance of culturomics, and/or direct quantification methods (e.g. qPCR), as a complementary approach for capturing rare resistance genes”. While we fully acknowledge that quantification of

absolute abundance would directly address this question, unfortunately, access to the original preterm samples - already limited in availability - is no longer possible.

o The lower microbial diversity in the PS group is likely due to the high proportion of supplemented bifidobacteria. However, it remains unclear whether this represents actual displacement of other species or a mere shift in relative abundance. The authors should also investigate microbial richness next to diversity and assess whether other bacterial taxa decrease in absolute terms.

We have now included the Chao1 index in Fig. 1b, Fig. 2g, and Supplementary Fig. 1, along with genus and species richness (i.e., absolute counts). Both the estimated richness (Chao1) and absolute richness (including singletons) yielded consistent results, suggesting that the changes reflect a displacement of taxa rather than a mere shift in relative abundance.

o The authors should consider providing read coverage statistics for pathobionts in PS vs. NPS datasets to confirm that the observed reduction is not a sequencing bias.

We used CoverM to estimate coverage of pathobionts including *Enterococcus*, *Staphylococcus*, *Klebsiella* and *Escherichia* (Supplementary Fig. 2) and all showed similar decreasing trends, confirming that the observed reduction was likely not due to sequencing bias.

ARG Detection and Biases

• The differences in ARG profiles could be influenced by hospital-specific bacterial communities rather than probiotics alone. The authors should acknowledge that circulating hospital microbiomes might contribute to the observed ARG trends.

We have since added a sentence in lines 404-406: “Nonetheless, we acknowledge the potential contribution of hospital-specific bacterial communities to the observed ARG profiles as evidenced by the high number of unique STs in each site.”

• The presence of *mcr-9* in samples collected from 2011–2012 is interesting but similar observations were also made for other *mcr* variants. Retrospective analyses both in silico on deposited metagenomes as well as on biobanked fecal samples or bacterial isolates have identified *mcr* genes in samples collected well before their first official reports. It could be stated that the current finding fits in a line of previous observations and highlights the value of the microbiome for AMR surveillance.

We have now added a sentence in lines 327-329 with references: “Our finding aligns with previous retrospective studies that identified *mcr-9* prior to its first official report, highlighting the value of metagenomics in AMR surveillance.”

• It remains unclear to the reviewer whether MDR characteristics were assessed only from metagenome-assembled genomes (MAGs) or if cultured isolates from the PS and NPS groups also showed differences. The latter would provide stronger evidence that MDR pathogens were actually reduced rather than simply undetected due to lower sequencing coverage.

We have examined both MAG and isolate genomes. We have now modified the sentence in lines 214-216: “Notably, none of the PS-associated *Klebsiella* (n=5) or *Escherichia* (n=12) genomes were MDR, while 47.6% of NPS-associated *Escherichia* (n=31) genomes, including cultured isolate genomes, exhibited MDR characteristics (Fig. 3e, bottom).” PS-linked *Escherichia* includes 5 isolate genomes, and NPS-linked *Escherichia* also has 5, within a total of 12 and 21 genomes (isolate + MAG) for PS and NPS respectively. These observations support our conclusion that MDR pathogens were genuinely reduced.

Multi-locus sequence typing

- The focus on *Enterococcus* HGT potential is valuable, but other species, but it would be interesting if the authors could also check whether the dominant sequence types (STs) of the other pathobionts differ between PS and NPS groups, as this could be indicative for environmental influences rather than probiotic-driven changes.

We have made minor adjustments to Fig. 3f to include pie charts showing the shared and unique STs. However, due to the small number of shared STs (majority are not shared STs), statistical testing is not possible.

Statistical Considerations and Figure Clarifications

- Multiple comparison corrections should be applied in differential abundance analyses (e.g., Figure 1g).

We performed multiple comparison corrections where possible and this now indicated in all figure legends.

- In Figure 1i, it would be helpful to specify which bacteria contribute to the unique differential pathways. Do these pathways primarily belong to bifidobacteria?

In Fig. 1i, the pathways are associated with a diverse range of bacterial taxa; however, they are more frequently linked to *Bifidobacterium* in the PS cohort, which is expected given that *Bifidobacterium* is a dominant taxon in this group. For clarity, we have provided a table showing pathway abundances (raw counts) alongside their corresponding bacterial taxa (Supplementary Table 4 and 5).

- Figure 2b should clarify what the number of ARGs represents (e.g., absolute counts, relative abundance?). Since higher bacterial diversity can naturally lead to more detected genes, including ARGs (Figure 2l), the authors might want to adjust for this in their analysis.

We made changes to y-axis label in Fig. 2b – it is now “distinct ARG count per infant”. In our analysis (Fig. 2l), higher bacterial diversity may lead to more detected genes, however the correlation was significant only in PS cohort with weak correlation. Therefore, this will not impact our conclusion.

Minor Comments

- Line 108 contains an unnecessary large space.

This is now rectified.

Reviewer #3 (Remarks to the Author):

References

- 1 Alcon-Giner, C. *et al.* Microbiota Supplementation with *Bifidobacterium* and *Lactobacillus* Modifies the Preterm Infant Gut Microbiota and Metabolome: An Observational Study. *Cell Rep Med* **1**, 100077 (2020).
<https://doi.org/10.1016/j.xcrm.2020.100077>

- 2 Brown, C. T., Olm, M. R., Thomas, B. C. & Banfield, J. F. Measurement of bacterial replication rates in microbial communities. *Nat Biotechnol* **34**, 1256-1263 (2016). <https://doi.org/10.1038/nbt.3704>

Reviewer #1 (Remarks to the Author):

The authors have satisfactorily addressed the comments regarding consistency of abundance measurements (1), caveats on the surprisingly minor impact of antibiotic treatment (2), and the presentation of time series data (5). For the comment on colonization of *Bifidobacterium* (3), I recognize that quantification is not possible in this case, and the evidence the authors present addresses the comment.

The authors have also added appropriate caveats on ARG detection in the metagenomic data (comment 4). However, while the species-level rarefaction analysis is a convincing argument that taxonomic detection is not a problem, I suggest that a gene- or mechanism- level rarefaction analysis would be more relevant to the ARG detection discussion (e.g. Zaheer et al. Sci Rep. 2018. <https://doi.org/10.1038/s41598-018-24280-8>). Further, is it the contribution of very low abundance ARGs that explains why the decreased ARG abundance observed in the PS metagenomes (Fig. 2b) is not observed in the PS isolates (Supplementary Fig. 5b,c)? Last, Figure 4h, which presents interesting and important information, remains visually confusing to decipher because of difficulty mapping the infant symbols on the right to rows in the “cohort”, “individual”, and “time” columns. If I’m reading the plot correctly, perhaps extending the boxes separating values in the “ANI” column is a solution.

We thank the reviewers for their thoughtful and constructive comments. Regarding the gene-level rarefaction analysis:

1. We agree that this type of analysis could help address the presence and contribution of low-abundance ARGs. However, we have opted not to include this analysis, as it falls outside the scope of the current study. Nevertheless, we have added discussion of this caveat in the revised manuscript, particularly in reference to ARGs identified in PS isolate genomes. These showed a trend consistent with our metagenomic findings - namely, reduced ARGs with probiotic supplementation - although this was not statistically significant (Supplementary Fig. 5b). We note, however, that this analysis was based on a limited subset of five infants per cohort, which we have clearly acknowledged as a limitation. We are looking forward to a future study specifically aimed at exploring the role and detection of low-abundance ARGs in the preterm gut (lines 420-421). This will be particularly important for informing optimal sequencing depth and methodological approaches in this population.
2. We would also like to highlight a recently published study (June 2025, *Nature Communications*) that reports similar findings in neonates. The authors observed a statistically significant reduction in ARGs in the gut microbiome associated with *Bifidobacterium* (<https://doi.org/10.1038/s41467-025-61154-w>). In addition, a study published in 2022 (<https://doi.org/10.1093/ajcn/nqab353>) reported that formula-fed infants carry a higher load of antibiotic resistance genes compared to those fed with human milk, primarily linked to a reduced abundance of *Bifidobacterium* in the formula-fed group (these two references are now added in lines 381-382). Therefore, we believe our conclusion remains robust and is unlikely to be affected by any undetected, rare antibiotic resistance genes (ARGs), if present.

Secondly, the question of whether low-abundance ARGs might explain the discrepancy between the decreased ARG abundance observed in metagenomes and the lack of this trend in PS isolates is a valid and insightful one. While our current data cannot definitively answer this, we agree that further investigation is warranted. However, based on our current analyses, we do not believe this limitation alters the primary conclusions of the study.

Additionally, while ultra-deep sequencing could help capture a broader range of certain rare ARGs, this approach is resource-intensive and currently beyond the scope and funding of our study. We note that our analysis suggests approximately 7.9% of lower-abundance ARGs may be missed in metagenomics but can be recovered through culturomics. However, as we did not perform extensive untargeted culturing in this study, we cannot draw firm conclusions from this observation.

Our aim was not to determine optimal sequencing depth, but rather to characterise ARG dynamics within a realistic and resource-constrained clinical context. The samples were sequenced to an average depth of 10 million reads per sample (circa 2018), which is generally considered sufficient for low-diversity preterm gut microbiomes, particularly in the first three weeks of life. This depth was selected to balance robust resolution with available resources, and we believe it was appropriate for the study objectives.

We fully agree with the reviewers that deeper investigations into low-abundance ARGs and sequencing strategies are important directions for future research.

Finally, we have modified **Fig. 4h** in the revised manuscript to improve clarity, as suggested.

Reviewer #2 (Remarks to the Author):

The authors have done extensive revisions, including additional analyses (e.g. on replication rate of *B. bifidum*), nuancing the discussion and highlighting the consistency with prior findings (e.g., mcr-9.1) and clarification and further elaboration (e.g., the likely limited impact of study site as major confounding factor). While it remains a limitation that quantitative approaches can no longer be conducted due to lack of remaining samples/DNA, the authors have at least excluded the role of mere dilution effects or sequencing depth bias as good as possible using computational approaches. All my comments and concerns have thereby been adequately addressed.

We are grateful for the reviewers' thoughtful comments, which have helped us improve the manuscript.

Reviewer #3 (Remarks to the Author):
